# A chaperonin complex regulates organelle proteostasis in malaria parasites

Amanda Tissawak[1,2], Yarden Rosin[1,2], Shirly Katz Galay[1,2], Alia Qasem[1,2], Michal Shahar[1,2], Nirit Trabelsi[1], Ora Furman-Schueler[1], Steven M. Johnson[3], Anat Florentin[1,2]*

1 Department of Microbiology and Molecular Genetics, Faculty of Medicine, The Hebrew University of Jerusalem, Jerusalem, Israel, 2 The Kuvin Center for the Study of Infectious and Tropical Diseases, Faculty of Medicine, The Hebrew University of Jerusalem, Jerusalem, Israel, 3 Department of Biochemistry and Molecular Biology, Indiana University School of Medicine, Indianapolis, Indiana, United States of America

* anat.florentin@mail.huji.ac.il

## Abstract

The apicoplast of *Plasmodium* parasites serves as a metabolic hub that synthesize essential biomolecules. Like other endosymbiotic organelles, 90% of the apicoplast proteome is encoded by the cell nucleus and transported to the organelle. Evidence suggests that the apicoplast has minimal control over the synthesis of its proteome and therefore it is unclear how organelle proteostasis is regulated. Here, we identified and investigated a large and conserved chaperonin (CPN) complex with a previously unknown function. Using genetic tools, we demonstrated that ablation of the apicoplast CPN60 subunit leads to parasite death due to organellar damage, immediately within its first replication cycle, deviating from the delayed death phenotype commonly observed for apicoplast translation inhibitors. Unlike its close orthologues in other prokaryotic and eukaryotic cells, CPN60 is not upregulated during heat shock (HS) and does not affect HS response in the parasite. Instead, we found that it is directly involved in proteostasis through interaction with the Clp (caseinolytic protease) proteolytic complex. We showed that CPN60 physically binds both the active and inactive forms of the Clp complex, and manipulates its stability. A computational structural model of a possible interaction between these two large complexes suggests a stable interface. Finally, we screened a panel of inhibitors for the bacterial CPN60 orthologue GroEL, to test the potential of chaperonin inhibition as antimalarial. These inhibitors demonstrated an anti-*Plasmodium* activity that was not restricted to apicoplast function, with additional targets outside of this organelle. Taken together, this work reveals how balanced activities of proteolysis and refolding safeguard the apicoplast proteome, and are essential for organelle biogenesis.

**Data availability statement:** All relevant data are in the manuscript and its supporting information files.

**Funding:** This work was supported by grants from the Israel Science Foundation [400/22 and 2786/22 to AF and 301/21 to OFS]. The funders had no role in study design, data collection and analysis, decision to publish, or preparation of the manuscript.

**Competing interests:** I have read the journal's policy and an author of this manuscript have the following competing interests: S.M.J. is a founder of BioEL Inc, which was formed to commercialize GroEL inhibitors for antibacterial applications.

## Author summary

The cell of the human malaria parasite *Plasmodium falciparum* has a unique organelle called the apicoplast that produces essential metabolites, but it is unclear how it maintains a stable proteome. Here, we address the question of organelle proteostasis by investigating the function of a large chaperonin complex and its main subunit CPN60. We show that CPN60 mutants die due to organellar damage immediately within the first replication cycle, avoiding the typical apicoplast-delayed cell death. We demonstrate that it binds and stabilizes another large proteolytic complex and use computational tools to predict how a stable interface is attained. We use bacterial inhibitors to explore their potential as an antimalarial drug target. This study reveals how balanced refolding and proteolysis safeguard the apicoplast proteome and opens a new avenue for antimalarial drug discovery.

## Introduction

The malaria parasite, *Plasmodium falciparum*, remains one of the deadliest human pathogens, causing 600,000 deaths yearly worldwide [1]. Limited vaccines efficacies and spreading of drug resistance are major global concerns. Despite years of research, many aspects of the parasite's basic cell biology remain obscure, hindering rational drug design and improved therapeutic interventions. Like most other related parasites from the phylum of *Apicomplexa*, this single-celled eukaryotic organism contains a non-photosynthetic plastid called apicoplast [2,3]. This unique organelle evolved via a two-step endosymbiosis; in the primary endosymbiotic event, a cyanobacterium was incorporated into a eukaryotic cell to form the modern chloroplast. During the secondary endosymbiotic event, a photosynthetic red alga was further taken up by a protist and led to the formation of a secondary plastid [4]. Although not photosynthetic, the apicoplast harbors essential prokaryotic metabolic pathways that are essential to the parasite throughout its complex life cycle, including fatty acid, Fe-S clusters, and isoprenoids biosynthesis [5]. The *Plasmodium* apicoplast shares molecular features with prokaryotes, plants and parasites, and therefore encompasses multiple parasite-specific drug targets. Indeed, drugs that target apicoplast biology are in clinical use [6]. Most of those drugs (e.g., doxycycline, clindamycin) target the prokaryotic protein synthesis machinery in the apicoplast [6,7]. However, less than 10% of the apicoplast proteome is encoded by its own genome. The vast majority of the hundreds of apicoplast proteins are encoded in the nuclear genome and are transported to the organelle via the secretory pathway [8,9]. Available transcriptomic data suggest that the apicoplast does not control the expression of these nuclear-encoded proteins [10]. In fact, parasites without an apicoplast continue to express and accumulate apicoplast proteins in vesicle-like structures in the cytoplasm [11–14]. Due to the inability of the apicoplast to control its own protein synthesis, it is unclear how the apicoplast maintains a stable proteome.

Previously, we suggested that protein degradation plays a central role in organelle proteostasis and showed that apicoplast-targeted Clp (caseinolytic protease) proteins form an essential proteolytic complex in the organelle [14,15]. The conserved Clp family consists of proteases and chaperones that form multi-subunit proteolytic complexes, with complex composition varying widely between different species and organelles [16,17]. In bacteria and plant chloroplasts, the ClpP proteases typically associate with Clp ATPase chaperones that unfold and feed substrates into the ClpP barrel-like cavity for degradation [18–20]. However, the putative *Plasmodium* Clp proteins differ significantly from their bacterial orthologs and their organellar, molecular and enzymatic functions are unclear [21,22]. Moreover, despite detailed genetic and molecular data illuminating complex composition, no regulator of the apicoplast Clp complex has been identified.

Besides the Clp chaperones, the *Plasmodium* apicoplast is predicted to host other molecular chaperones, among those, chaperonin 60. This 60 kDa Heat Shock protein, often termed HSP60 or CPN60, is highly conserved and found in nearly all organisms [23]. Its bacterial orthologue, GroEL, is involved in maintaining proteostasis under different stressors [24], and in most eukaryotic cells, *Plasmodium* included, a mitochondrial orthologue is required for organellar protein import [25]. In apicomplexan parasites, an additional CPN60 homolog was reported to localize to the apicoplast [15,26], although its functions are unknown.

Here, we report the molecular investigation of a non-mitochondrial CPN60 orthologue in the malaria parasite *P. falciparum*. We show that it localizes to the apicoplast and is primarily expressed during the late stages of the parasite's life cycle, when organelle biogenesis occurs. We found that CPN60 function is essential for parasites viability, but unlike its orthologs in most organisms, it is not involved in heat shock response. Instead, it is required for apicoplast functions and interacts with the proteolytic Clp complex. We further demonstrate that it binds the proteolytic Clp subunit and this interaction is required for Clp complex stability. Lastly, we screened a panel of inhibitors of the bacterial HSP60 orthologue GroEL in order to test the potential of chaperonin inhibition as antimalarial. Taken together, our studies shed light on the activity of the apicoplast CPN60 and its central role in maintaining organelle proteostasis and parasite's viability.

## Results

### *Plasmodium falciparum* chaperonin 60 is an apicoplast-localized protein

Our previous work suggested that the apicoplast employs caseinolytic protease (Clp)-mediated degradation to maintain organellar functions [14,15]. In order to further investigate proteostasis regulation in the apicoplast, we performed an affinity screen, immunoprecipitating several different Clp subunits coupled with mass spectrometry analysis [15]. Notably, we found a conserved apicoplast-resident heat shock protein (HSP60) homolog, termed chaperonin 60 (CPN60, PF3D7_1232100). The putative apicoplast CPN60 in *P. falciparum* is 718 amino acids in length, while the mitochondrial HSP60 (PF3D7_1015600) is 580 amino acids long, and the two isoforms share 30% sequence identity (46% similarity) (S1 Fig). To test whether and how CPN60 is involved in proteostasis and other apicoplast functions, we employed a genetic approach using the tetR-Dozi system, a *Plasmodium*-specific method [27] (Fig 1A). Using CRISPR/Cas9, we incorporated a V5 tag followed by the regulatory cassette at the 3′ end of the CPN60 gene, creating the CPN60^V5-apt parasite line. Genotyping of independently isolated clones was performed by PCR analysis, and showed correct integration into the *cpn60* locus (Fig 1B). Western blot analysis revealed expression of endogenous V5-tagged CPN60 at the predicted 85 kDa size (Fig 1C). The presence of a double band suggested that CPN60 contains an apicoplast-targeting transit peptide that is cleaved upon its organellar localization. To verify this, we treated CPN60^V5-apt parasites with the antibiotic chloramphenicol (CHL), which specifically inhibits apicoplast protein translation and leads to organelle loss [6]. 96-h post CHL treatment, we observed accumulation of the higher molecular weight band of CPN60 and disappearance of its lower molecular weight form (Fig 1C). This indicates that CPN60 has a transit peptide that cannot be cleaved off when organelle functions are compromised. Homologs of the HSP60 family are often considered house keeping genes, retaining constant expression levels throughout the cell cycle [23]. However, protein extraction from synchronized parasites revealed that CPN60 levels were relatively low at the first half of the 48-h asexual cycle, and increased only at the later trophozoite stage, 36

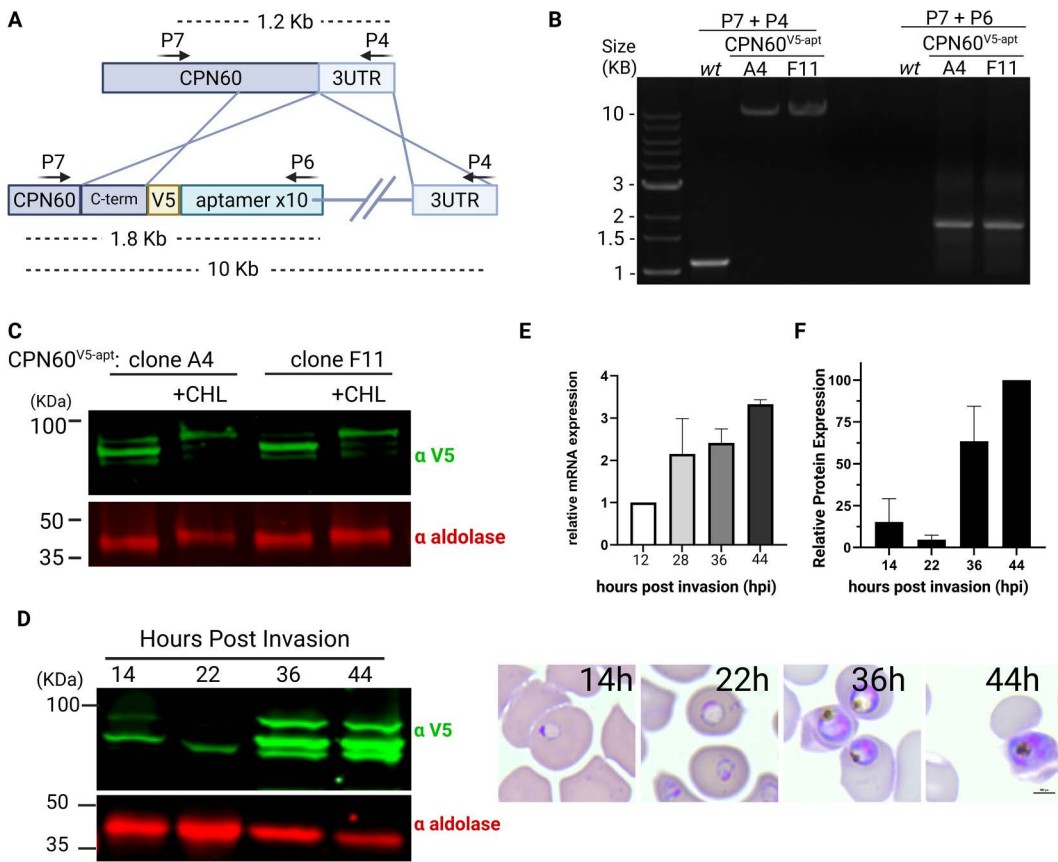

**Fig 1. Generation and characterization of transgenic CPN60$^{V5-apt}$ parasites.** A. Schematic representation of CRISPR-mediated homologous recombination through CPN60 homology regions enabling the integration of the V5 tag and the tetR-Dozi system. B. PCR amplification of genomic DNA isolated from CPN60$^{V5-apt}$ transfected parasites. Primers used are depicted in panel A and in Table 1. Primers are integration specific and thus can amplify 1.6 KB (P7+P6) and 11 KB (P7+P4) products only if integration occurs at the desired genomic locus. C. Western blot depicting two clones of CPN60$^{V5-apt}$ parasites (A4 and F11) following 96-hour incubation with or without 30 µM chloramphenicol (CHL) using an anti-V5 antibody and anti-aldolase as a loading control. D. CPN60$^{V5-apt}$ parasites were synchronized by incubation with 5% sorbitol followed by Percoll-mediated schizont-enrichment the following day. Western blot depicting expression profile of tagged CPN60 throughout parasite life cycle using an anti-V5 antibody. Right: Blood smears demonstrate parasites' developmental stage at the time of protein extraction. E. CPN60$^{V5-apt}$ parasite clones (A4 and F11) were synchronized using 5% sorbitol, and 8ml of culture were collected at the following time points: early ring stage (~12 h post-invasion), early trophozoite stage (~28 hpi), late trophozoite stage (~36 hpi), and schizont stage (~44 hpi). Parasites were liberated from host red blood cells using 0.1% saponin, and total mRNA was extracted and reverse-transcribed to cDNA. Quantitative real-time PCR (qRT-PCR) was performed to quantify transcript levels of the apicoplast-encoded chaperonin CPN60. Primers P14+P15 were used to amplify CPN60 and two housekeeping genes (aldolase P16+P17 and arg-tRNA synthetase P18+P19) for normalization. Gene expression was normalized to the mean ΔCt of both housekeeping genes (aldolase and arg-tRNA synthetase), and expression levels were calculated relative to the early ring stage using the ΔΔCt method. Data are presented as fold change (2^–ΔΔCt). Values represent the mean expression from two parasite lines, and error bars indicate the standard deviation. Statistical analyses were performed using GraphPad Prism. F. Densitometric analysis of protein bands from Western blot in 1D. V5 signal was normalized to aldolase. Figure created by Biorender.

hours post-invasion (Fig 1D, see protein levels and corresponding blood smears). This was also manifested with increased CPN60 mRNA levels, as measured by quantitative real-time PCR (Fig 1E). Quantification of CPN60 protein levels shows that expression peaks around the time that schizogony (nuclear replication), and apicoplast development begin [28] (Fig 1F).

We further monitored CPN60 expression and localization in different developmental stages using immunofluorescence assay (IFA). Recently, we showed that apicoplast development is tightly coordinated with nuclear replication and the cell cycle [29]. In accordance with these observations, we saw CPN60 at an early stage in a dot-like structure, in a typical elongated apicoplast morphology during the late trophozoite stage, a branched structure during schizogony and divided

daughter organelles post segmentation, in the last hour before egress (Fig 2A). We took advantage of ultrastructure expansion microscopy to verify the details of these structures and observed CPN60 in a consistent and uniform organelle localization (Fig 2B). Lastly, we combined this IFA with a mitotracker staining to demonstrate that the intricate structure is distinct from the parasite's mitochondrion and therefore marks the apicoplast (Fig 2C).

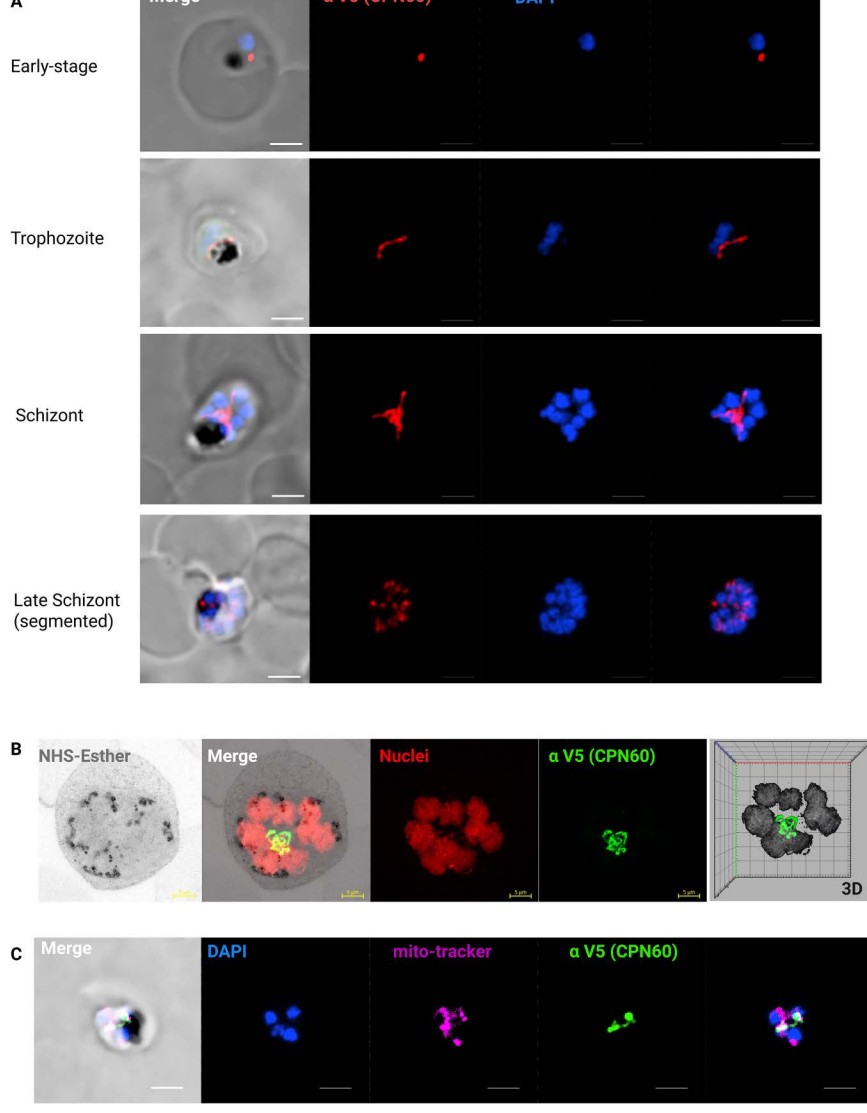

**Fig 2. Organellar Localization and expression of CPN60.** A. Immunofluorescence microscopy of CPN60[V5-apt] parasites in different developmental stages. Z stack images processed as Maximum Intensity show from left to right: merged DIC contrast, anti-V5 (CPN60, red), DAPI (parasite nucleus, blue), and merge of fluorescent channels. The imaging was performed using a Nikon Spinning Disk confocal fluorescence microscope equipped with a 1005 x/1.4NA objective. Scale bar is 2.5 μM. B. A representative image of an expanded parasite demonstrates uniform apicoplast localization of CPN60 in a well-developed organelle during mid-schizogony (eight nuclei). Images from left: NHS-Esther (total protein, grey), merge, DNA (red), anti-V5 antibody (CPN60, green), and a 3D visualization. Images were captured as Z-stacks (20 slices of 0.1 μm each) at 63x objective using Airyscan microscopy. The images are displayed as maximum intensity projections. Expansion factor is 5X. Scale bar is 5 μm. C. Immunofluorescence microscopy of CPN60[V5-apt] parasites processed as Maximum Intensity show from left to right: merged DIC contrast, DAPI (parasite nucleus, blue), MitoTracker Deep Red FM (mitochondrion, magenta), anti-V5 antibody (CPN60, green), and merge of fluorescentt channels. The imaging was performed using a Nikon Spinning Disk confocal fluorescence microscope equipped with a 100x/1.4NA objective. Scale bar is 2.5 μM.

## CPN60 is essential for parasite's survival due to its function in apicoplast biogenesis

To look into CPN60 cellular function, we activated the tet-aptamer system, to induce translational repression of CPN60 (Fig 3A). We induced protein knockdown by removal of the stabilizing ligand anhydrotetracycline (aTC), effectively reducing CPN60 protein levels within 24 hours, dropping below detection after 72 hours (Fig 3B). The consequences of CPN60 knockdown were parasite's death, indicating that its activity is essential for parasite's viability (Fig 3C). It was shown that apicoplast-related cytotoxicity can be reversed by addition of isopentenyl pyrophosphate (IPP), an essential apicoplast-producing metabolite [11]. We observed that media supplementation with IPP completely restored mutants' growth, indicating that CPN60 essential activity is restricted to apicoplast function (Fig 3D). Delayed cell death due to apicoplast-associated damage is observed in parasites treated with doxycycline or chloramphenicol. Such parasites complete their first 48 hours cell cycle, reinvade new cells and die only at the end of the 2nd cycle after 96 hours of treatment [30]. We therefore synchronized the cultures and removed aTC from early rings stage CPN60V5-apt parasites and observed their intraerythrocytic development (Fig 3E). During the first cycle, knockdown parasites developed like +aTC control until the trophozoite stage, but then could not process into normal schizogony (Fig 3F). The few persisting parasites in the culture could not egress or reinvade and failed to form new ring-stage parasites in the 2nd cycle (Fig 3G). For comparison, we induced knockdown in a different apicoplast-resident protein, PfClpS, which we previously showed to be required for organelle biogenesis [15]. Induction of PfClpS knockdown using the same tet-aptamer system under the same experimental settings resulted in a delayed cell death, with parasites death observed 5 days post aTC removal (S2 Fig). This reinforces the observation that, unlike many other apicoplast proteins, CPN60 ablation results in an apicoplast-immediate cell death.

We then tested whether essentiality of CPN60 is limited to supporting an apicoplast metabolic pathway, or it is also required for the physical biogenesis of the organelle. If the former is correct, then IPP supplementation during CPN60 knockdown should be sufficient to retain an intact organelle, albeit dysfunctional. However, if the latter is true and CPN60 is required for apicoplast biogenesis, its knockdown will result in organelle loss, regardless of IPP supplementation. To distinguish between these two possibilities, we grew CPN60V5-apt parasites for 17 days without aTC while supplementing the media with IPP, and then added back aTC to restore CPN60 expression (Fig 3H). We observed that this treatment resulted in parasites growth that is completely dependent on IPP supplementation, regardless of aTC presence (i.e., CPN60 expression) (Fig 3H). Moreover, IFA showed that CPN60 in treated parasites lost its distinct apicoplast elongated structure and appeared instead in vesicle-like cytoplasmic puncta (Fig 3I). Moreover, processing of CPN60 was lost and it migrated on the gel at the size of its cytoplasmic form (Fig 3J). Collectively, these results demonstrate that CPN60 activity is required for apicoplast biogenesis, and during knockdown the organelle is lost and cannot be retrieved even when CPN60 expression is restored.

## Plasmodium CPN60 is not involved in heat shock response

In most organisms, HSP60 family members are involved in stress pathways, particularly heat shock (HS) response. During infection, the malaria parasite faces periodic fevers, making HS one of its main stressors. To investigate the function of apicoplast CPN60 during HS, we measured its expression levels following 24 hours incubation at 40°C. *Wildtype* parasites subjected to HS, grew slower and exhibited morphological defects (Fig 4A). However, this response was not accompanied by elevated CPN60 protein levels (Fig 4B). We also followed potential changes in transcript levels following HS, and as a positive control used leucine-rich repeat protein (LRR5), a *Plasmodium* gene that is nonessential under ideal growing conditions, however was found to be associated with HS response in the parasite [31,32]. Here too, we found that while LRR5 transcripts were elevated by two- and three-folds post HS, there was no effect on CPN60 transcription (Fig 4C).

Next, we sought to analyze the effect of CPN60 depletion on parasite's survival following HS. Because complete knockdown of CPN60 leads to parasite's death, we were interested in decreasing CPN60 levels to the point where subtle effects could be tested. To do that, we performed a half-maximal effective concentration (EC50) assay for aTC, and found

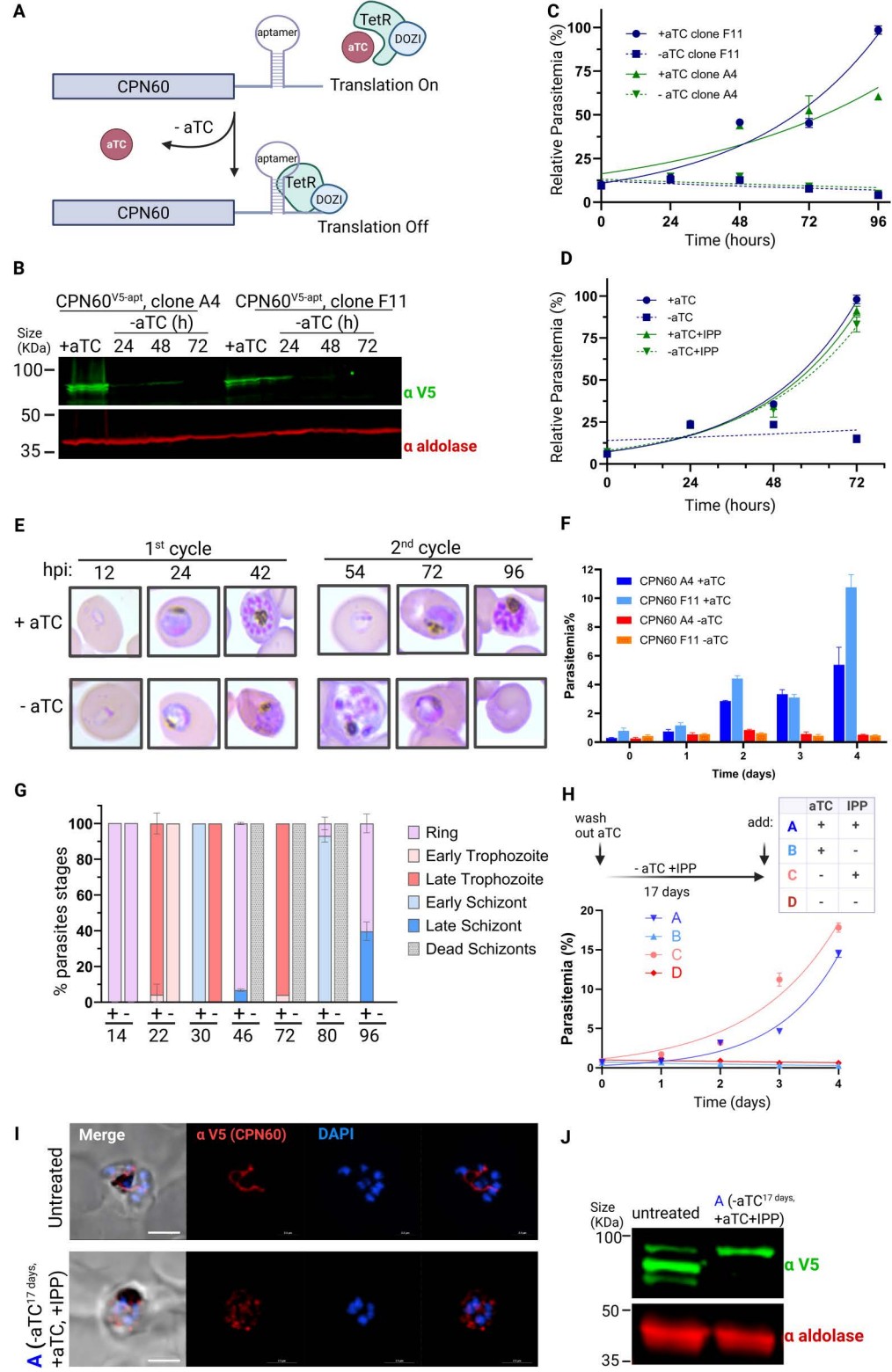

**Fig 3. CPN60 is essential for parasite's viability due to its apicoplast function.** A. Schematic representation of the tetR-Dozi system, which enables protein knockdown upon removal of the tetracycline analogue aTC. B. Western blots depicting the expression of the tagged CPN60 using an anti-V5 antibody following knockdown induction. A significant reduction in protein levels is already observed after 24 hours, and falls below detection 72 hours

after aTC removal. C. Growth curve of CPN60^V5-apt parasites. Independent CPN60^V5-apt clones A4 and F11 were grown with or without 0.5 mM aTC, and parasitemia was monitored every 24 h over two cycles via flow cytometry. The 100% of growth represents the highest value of calculated parasitemia (final parasitemia in the presence of aTC). Normalized data are represented as mean±SEM for three technical replicates. D. Growth curve of CPN60^V5-apt parasites, clone A4, supplemented with 200 μM IPP. The 100% of growth represents the highest value of calculated parasitemia (final parasitemia in the presence of aTC). Normalized data are represented as mean±SEM for three technical replicates. E. CPN60^V5-apt parasites were synchronized by incubation with 5% sorbitol followed by Percoll-mediated schizont-enrichment the following day. Synched parasites were washed 8 times and then incubated with or without aTC. Giemsa-stained blood smears were imaged using an upright Eclipse E200 Microscope. F. Growth rates of synchronous CPN60^V5-apt parasites. Clones A4 and F11 were grown with or without 0.5 mM aTC, and parasitemia was monitored every 24 h via flow cytometry. Data are represented in columns as mean±SEM for three technical replicates. G. Blood smears of synched CPN60^V5-apt parasites with or without aTC were analyzed for specific parasites developmental stages. Minus aTC parasites demonstrate a delayed trophozoite stage already on the first replication cycle, fail to egress, and die during schizogony. Dead schizonts are defined as morphologically aberrant mutant cells with more than one nucleus that linger for more than 48 hours but did not complete their first replication cycle. H. Parasites were cultured without aTC and supplemented with IPP for 17 days. Parasites culture was then divided into four groups treated as following: (A) +aTC,+IPP; (B) +aTC, -IPP; (C) -aTC,+IPP; (D) -aTC, -IPP. Parasitemia of these four groups was monitored every 24 hours over two cycles via flow cytometry. Normalized data are represented as mean±SEM for three technical replicates. I. Immunofluorescence microscopy of untreated and group (A) CPN60^V5-apt parasites. Z stack images processed as Maximum Intensity show from left to right: merged DIC contrast, anti-V5 (CPN60, red), DAPI (parasite nucleus, blue), and merge of fluorescentt channels. The imaging was performed using a Nikon Spinning Disk confocal fluorescence microscope equipped with a 100x/1.4NA objective. Scale bar is 2.5 μM. J. Western blot depicting expression of untreated and group (A) CPN60^V5-apt parasites using an anti-V5 antibody and anti-aldolase as a loading control.

that 4.5 nM support 50% survival under normal temperature as well as following incubation at 40°C (Fig 4D). Next, we measured CPN60 protein levels while growing the parasite's with 2X EC50 dose (8 nM aTC) compared to its levels in non-induced parasites which are cultured at 100X EC50 (500 nM aTC, Fig 4E). Quantification revealed that at this aTC concentration, CPN60 cellular levels were decreased by 50% (Fig 4E). Next, we measured parasite's growth at different aTC concentrations following 24 hours incubation at 40°C. While *wildtype* parasites recover after heat exposure, their growth is slightly attenuated (Fig 4F, blue curve). Importantly, parasites with reduced CPN60 levels (16, 8, 4 nM which correspond to purple, green and red curves, respectively in Fig 4F) exhibited compromised growth, in an aTC dose-dependent manner. However, exposure to heat did not alter their relative growth rate compared to the effect that it had on *wildtype* parasites (Figs 4F and S2B). These data suggest that unlike canonical CPN60 orthologues, the *Plasmodium* CPN60 is probably not involved in HS response and possesses an apicoplast-specific function.

## Co-expression and co-localization of CPN60 and PfClpP, the active protease of the Clp complex

We previously identified CPN60 in a mass spectrometry screen as a potential interactor of the Clp complex [15]. To look deeper into this potential interaction, we expressed a tagged copy of the Clp protease subunit, PfClpP. PfClpP is a highly potent protease and in order to avoid adverse effects, we used a transgenic PfClpP with a point mutation in its proteolytic active site (S264A), rendering it catalytically inactive [15]. Furthermore, we designed integration into the parasite's genome under the endogenous *hsp110* promoter, which enables constant but moderate levels of expression [15]. This construct was transfected on the background of the CPN60 conditional mutants, generating the CPN60^V5-apt; ClpP^DEAD-Ty parasite line (Fig 5A). Clones were isolated and integration at the *hsp110* locus was verified using diagnostic PCR (S3A Fig). Western blot demonstrated the co-expression of the endogenous CPN60^V5 at 85 KDa, as well as the transgenic ClpP^DEAD-Ty (Fig 5B). The expected processing pattern of PfClpP is consistent with a model of a full-length PfClpP, presumably in transit through the secretory pathway (fraction III), an apicoplast-localized PfClpP (fraction II), and a mature PfClpP after proteolytic removal of an inhibitory pro-domain (fraction I) (Fig 5B). We verified that expression of ClpP^DEAD-Ty had no effect on parasite's growth or CPN60 function, as can be seen by a consistent EC50 at 37°C or following HS (Fig 5C). Immunofluorescent microscopy, revealed that CPN60 co-localizes with PfClpP in the apicoplast during all cell cycle stages (Fig 5D). Therefore, we concluded that ectopic expression of ClpP^DEAD-Ty does not affect its processing or overall apicoplast morphology. Furthermore, it colocalizes with CPN60 without affecting its activity, indicating that the co-expression could be used to evaluate interaction between the two.

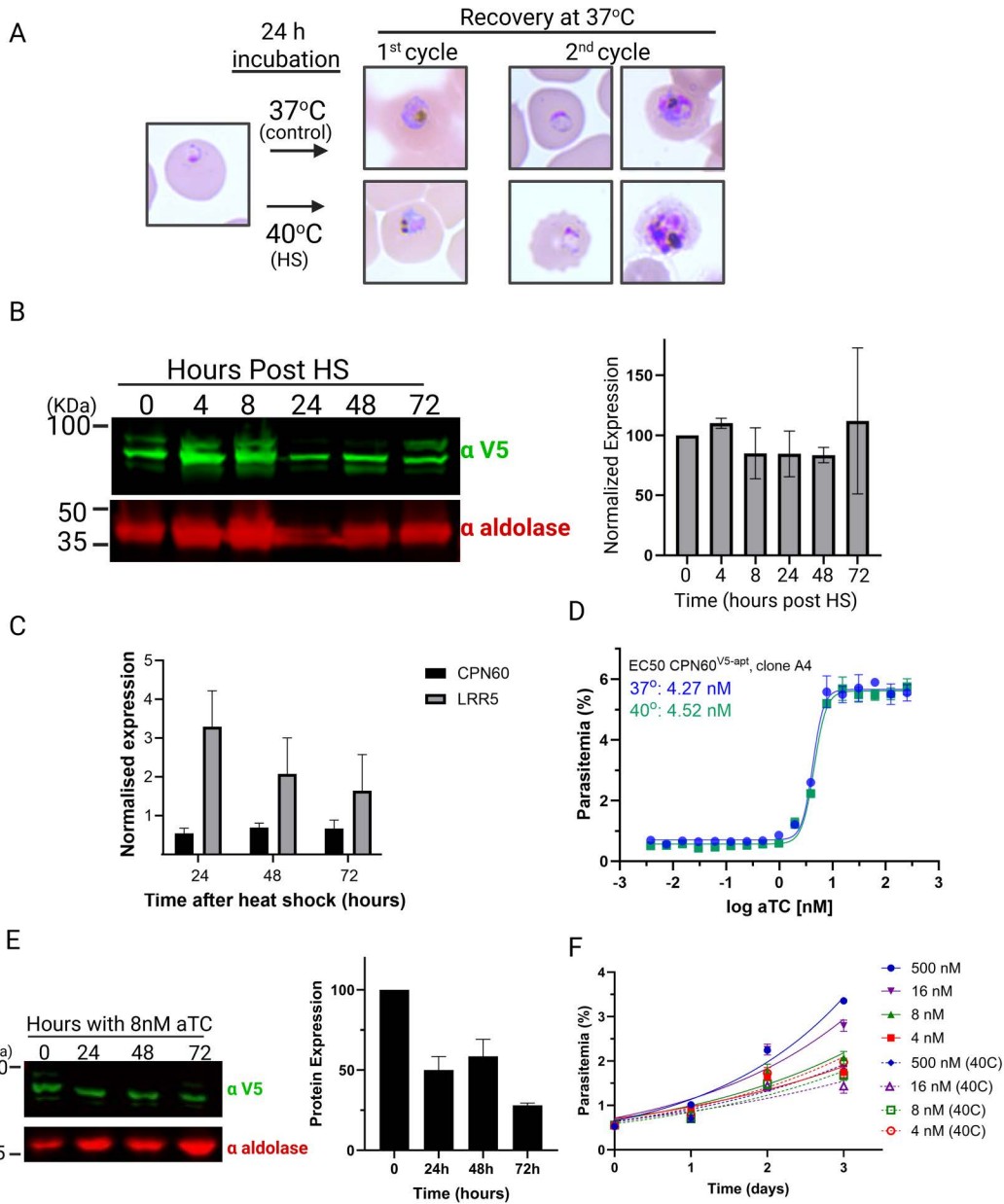

**Fig 4. CPN60 is not involved in Heat Shock response.** A. Workflow of HS experiments and representative blood smears. Parasites are subjected to HS at 40°C for 24 hours and then allowed to recover at normal temperature (37°C). B. Parasites were subjected to HS and recovery as depicted in (A). Left: Cellular lysates were extracted every 24 hours over a span of four days and subjected to Western blot analysis using an anti-V5 antibody to follow tagged CPN60 levels. Right: Quantification of CPN60 levels post-heat shock demonstrate no significant increase in protein levels. C. *P. falciparum* culture was subjected to heat shock at 40°C for 24 hours, followed by return to standard conditions (37°C). Twelve milliliters of parasite culture were collected at 24, 48, and 72 hours post-heat shock for RNA extraction and cDNA synthesis. Quantitative real-time PCR (qRT-PCR) was performed to assess the expression levels of CPN60 (P14+P15), and the heat shock-responsive gene LRR5 (P20+P21). Gene expression was normalized to the mean ΔCt of two housekeeping genes; aldolase (P16+P17) and arg-tRNA synthetase (P18+P19), and values were calculated relative to the non-heat-shocked control using the ΔΔCt method. The graph displays −ΔΔCt values. Error bars represent the standard deviation from three biological replicates. While LRR5 expression increased post-heat shock, CPN60 levels remained unchanged, suggesting that CPN60 is not transcriptionally regulated in response to heat stress. D. To calculate EC50 of aTC, CPN60$^{V5-apt}$ parasites were thoroughly washed to remove aTC, and then incubated in serial aTC dilutions in a 96-well plate. Parasitemia was measured after 4 days using flow cytometry, showing an EC50 of 4.3 nM at 37°c or 4.5 nM following 24 h incubation at

40°C. Data are fit to a dose-response equation and are represented as mean ± SEM. One representative experiment out of three is shown. E. Parasites were thoroughly washed and then incubated with 8 nM aTC (2X EC50). Left: Parasites lysates were obtained every 24 hours and subjected to western blot analyses by being probed with antibodies against V5 (green) and aldolase (loading control, red). The protein marker sizes that co-migrated with the probed protein are shown on the left. Right: Densitometric analysis of protein bands on the left indicating 50% drop in CPN60 expression. F. CPN60$^{V5-apt}$ parasites were washed and incubated with different aTC concentrations (8 nM, 4 nM, 16 nM and 500 nM), subjected to HS and then allowed to grow at 37°C for three days, while being measured daily by flow cytometry. Data are fit to an exponential growth curve and are represented as mean ± SEM for three technical replicates.

## CPN60 binds and stabilizes the Clp complex

To test for physical interactions between CPN60 and PfClpP, we performed immunoprecipitation using the double-mutant parasites. The pulldown of ClpP$^{DEAD-Ty}$ resulted in the co-IP of CPN60, in the form of a weak band (Fig 5E, right panel). The reciprocal IP of CPN60 revealed that it interacts with the mature ClpP protease (I), as well as with the zymogen form (II) but not with the full length PfClpP fraction (III) (Fig 5E, left panel). Importantly, pulldown of CPN60$^{V5}$ was specific to PfClpP as it failed to co-IP unrelated highly-expressed proteins (S3B Fig), suggesting a specific interaction with ClpP however weak or transient. This indicates that CPN60 and PfClpP interact only upon co-localization to the apicoplast and that the chaperone binds the Clp complex both before and after protease activation. This might be a direct interaction, or indirect through a third unknown partner.

Typically, Clp proteolytic subunits oligomerize into heptameric rings, consisting of the core of the Clp complex [17]. Likewise, CPN60 orthologues are part of large heptameric or tetradecameric chaperonin complexes [23]. Therefore, the interaction observed here is unlikely to occur between individual monomers, rather at an interface between large complexes. To better understand this interaction, we utilized previously solved structures of CPN60 (PDB ID 7K3Z) [33] and PfClpP (PDB ID 2F6I) [22] and set to predict how these two large complexes might interact in their 3D conformations (S3C and S3D Figs). We used AlphaFold3 (AF3) [34], a state-of-the-art model with a diffusion-based architecture capable of handling large protein complexes, offering improved accuracy and speed compared to AlphaFold2 [35]. The AF3 predictions provided a model in which the two rings are stacked on each other, whereby the individual monomers tilt in opposite directions. Each monomer of the PfClpP heptameric ring makes contacts with two monomers in CPN60, and each CPN60 monomer contacts two monomers in PfClpP (S3E Fig). In this configuration it is possible for both CPN60 and ClpP complexes to form tetradecameric barrels, which interact via a stable interface.

We next examined the significance of this interaction, and tested possible effects on PfClpP activity. We previously showed that PfClpP accumulates in the apicoplast as an inactive zymogen, and upon assembly of the PfClpP oligomer, it is auto-cleaved to remove the inhibitory pro-domain and becomes activated [15]. Thus, the ratio between the inactive (band II) and active (band I) PfClpP forms is indicative of its activation state. To test the processing pattern of PfClpP upon CPN60 knockdown, we removed aTC from non-synchronous parasites and observed PfClpP processing 24 hours later, before parasites begin to die. We detected a reduction in the levels of PfClpP active band (I) and accumulation of the inactive zymogen (II), once CPN60 levels drop (Fig 5F). This shift in the ratio of active: inactive PfClpP could be quantified (Fig 5G), suggesting that the interaction between CPN60 and PfClpP is essential for PfClpP activation and complex stability.

## Exploiting the apicoplast chaperonin system as an antimalarial target

Inhibition of protein synthesis in the apicoplast is a clinically approved drug target, as indicated by the antimalarial effect of doxycycline. Nevertheless, such treatment directly affects only a fraction of the total apicoplast proteome. Although largely unexplored, post-translational regulation in the organelle is no less critical and likewise can prove druggable. In this regard, the essentiality of CPN60 described here supports its viability as a potential antimalarial target.

To test this, we took advantage of a series of bacterial inhibitors designed to target the prokaryotic CPN60 orthologue, GroEL. As far as we are aware, chaperonin inhibitors have never been explored against *Plasmodium*. Thus, we conducted a pilot study to test the efficacy of a panel of phenylbenzoxazole (PBZ) analogues. PBZ compounds

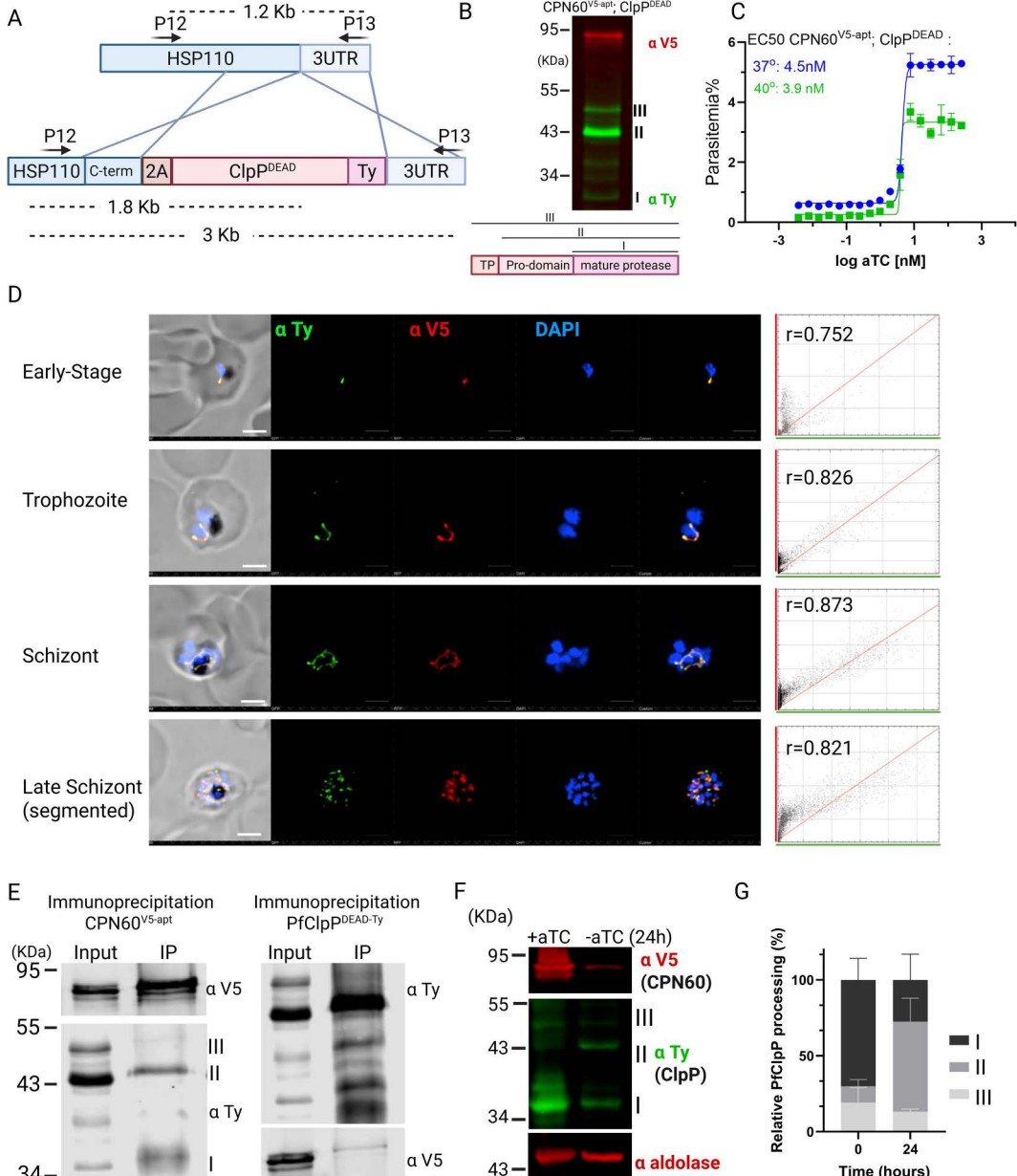

**Fig 5. CPN60 co-localizes, binds and stabilizes PfClpP.** A. A diagram depicting integration of a transgenic inactive PfClpP with a point mutation in the proteolytic active site (S264A) and a C-terminal Ty tag. CRISPR-mediated integration into the parasite's genome under the endogenous *hsp110* promoter enables constant and moderate expression. B. Western blot analysis of lysates from CPN60$^{V5\text{-apt}}$; ClpP$^{DEAD\text{-Ty}}$ parasite line, probed with antibodies against V5 (red) and Ty (green) showing the co-expression of the two proteins. Bottom: Schematic representation of the PfClpP processing: Full length cytoplasmic PfClpP with the transit peptide (TP, III), apicoplast-localized fraction after TP removal (II) and mature PfClpP after proteolytic cleavage of the pro-domain (I). C. EC50 of aTC in CPN60$^{V5\text{-apt}}$; ClpP$^{DEAD\text{-Ty}}$ parasite in 37°C (blue) and following 24 hours incubation at 40°C (green). Parasites were washed to remove aTC, and then incubated in serial aTC dilutions in a 96-well plate. Parasitemia was measured after 4 days using flow cytometry, showing comparable EC50s of 4.5 nM and 3.9 nM. Data are fit to a dose-response equation and are represented as mean ± SEM. One representative experiment out of three is shown. D. Immunofluorescence microscopy of CPN60$^{V5\text{-apt}}$; ClpP$^{DEAD\text{-Ty}}$ in different developmental stages. Z stack images processed as Maximum Intensity show from left to right: merged fluorescentt channels and DIC contrast, anti-Ty (ClpP$^{DEAD\text{-Ty}}$, green), anti-V5 (CPN60, red), and DAPI (parasite nucleus, blue), along with merged fluorescent channels. The imaging was performed using a Nikon Spinning Disk confocal fluorescence microscope equipped with a 100x/1.4NA objective. Scale bar is 2.5 µM. Right: Cytofluorograms depicting the fluorescence of each pixel in green channel (x-axis) against the red channel (y-axis) of maximum projection confocal images on the left. Pearson's coefficients (r) above 0.5 indicate significant co-localization between two fluorescent channels. E. Co-IP of CPN60$^{V5\text{-apt}}$ and ClpP$^{DEAD\text{-Ty}}$. Parasites were isolated and sonicated, and extracts

were incubated with either anti-V5 antibody-conjugated beads (for CPN60 pulldown, left) or anti-Ty antibody-conjugated beads (for PfClpP pulldown, right). Input and IP samples were loaded on SDS-page and blotted with anti-Ty and anti-V5 antibodies. F. CPN60$^{V5-apt}$; ClpP$^{DEAD-Ty}$ parasites were incubated without aTC for 24 h, and lysates were extracted and subjected to Western blot analysis using antibodies against V5 (CPN60, red), Ty (ClpP$^{DEAD-Ty}$ parasite, green) and aldolase (red). CPN60 knockdown results in the disappearance of the processed form of PfClpP (I) and accumulation of the inactive zymogen (II). Gel image is representative of three biological replicates. G. Densitometric analysis of PfClpP protein bands from Western blot in (F). Each PfClpP processed form (I, II, III) was calculated as a fraction of total PfClpP expression. Results indicate a significant reduction in PfClpP processing upon CPN60 knockdown, suggesting that the interaction with CPN60 is essential for PfClp complex activity.

were previously shown to bind and inhibit the activity of GroEL in *E. coli*, inducing a separation and a twist of the two rings that decoupled their inter-ring allosteric signaling, thereby blocking the folding cycle [36]. Detailed cryoEM structures showed that the PBZ analogue PBZ1587 binds at the GroEL ring-ring interface [37]. Moreover, it was shown that mutations of key residues within the interface binding site render GroEL resistance both *in vitro* and in *E. coli*, demonstrating on-target antibacterial effects and a validated tool for other studies. We therefore hypothesized that PBZ analogues have the potential to bind the *Plasmodium* chaperonin systems, either the apicoplast CPN60, the mitochondrial HSP60, or both. Sequence alignments indicated that *E. coli* GroEL, human mtHSP60, and *P. falciparum* mitochondrial HSP60 shared high conservation of their overall sequences and the PBZ1587 ring-ring interface binding sites (S1 and S4 Figs), but apicoplast CPN60 was much less conserved. However, upon generating homology models of apicoplast CPN60 and mitochondrial HSP60 bound with PBZ1587 at the ring-ring interface as it does with GroEL, we were encouraged to see that, despite the differences in binding site residues, many key ligand-binding site interactions were predicted (S5 and S6 Figs), suggesting the inhibitors could still be effective in *P. falciparum*. Furthermore, the apicoplast CPN60 isoform contains an extended alpha-helix at the ring-ring interface near the PBZ1587 binding site, a feature we hoped might provide selective targeting as the site is more well defined compared with mitochondrial HSP60.

To test the antimalarial potential of the PBZ inhibitors, we incubated parasites with serial dilutions of each (50 µM to 50 nM) and monitored growth after 72 or 96 hours to calculate the inhibitor EC50 (Figs 6 and 7A). Although some inhibitors showed moderate activity against *P. falciparum* growth (1134, 1135, 1259), others produced an EC50 at a single-digit micromolar range (1001, 1064, 1133, 1207, 1221) or lower (1587), demonstrating a potent anti-*Plasmodium* activity (Figs 6 and 7A). We then incubated synchronized ring-stage parasites with a selected panel of the more potent inhibitors and followed their intraerythrocytic replication. During the first cycle, treated parasites developed normally like untreated control until the trophozoite stage, but then could not proceed to normal schizogony, and failed to complete their cycle (Fig 7B). The kinetics of this developmental failure was comparable to the one observed upon CPN60 knockdown (Fig 3G). We therefore tested whether inhibitors cytotoxicity is associated with impaired apicoplast development. We incubated non-synched CPN60$^{V5-apt}$ parasites with PBZ inhibitors, and fixed the cells after 24 hours, before parasites begin to die. Strikingly, IFA demonstrated that PBZ treatments led to aberrant apicoplast morphologies, including fragmented, shrunk, or undeveloped organelles (Fig 7C). These results suggest at least one apicoplast-associated target, with CPN60 being a likely candidate.

We tried to test this directly, and performed EC50 assays for the PBZ compounds during CPN60 partial knockdown. This, however, did not lead to a significant EC50 shift for the compounds, hindering definitive conclusions (S7 Fig). We therefore assessed specificity towards broader apicoplast function, by coincubation of PBZ compounds with IPP. We incubated the parasites with a fixed inhibitor concentration of 12.5 µM (roughly 2-3X of most EC50s), and monitored their growth over 96 hours. While IPP did not affect growth of untreated parasites, it fully restores viability of chloramphenicol-treated parasites, which inhibit apicoplast translation (Fig 8A). However, the toxic effect of the PBZ inhibitors could not be restored by IPP, suggesting that (1) the 12.5 µM test concentration may have been too high above inhibitor EC50 values, impeding rescue by IPP supplementation (i.e., PBZ1587), or (2) inhibitors could be functioning against additional targets, the most likely candidate being mitochondrial CPN60. However, the moderately potent inhibitor PBZ1214 (EC50 = 9.6-12.5 µM)

| Test PBZ # | Structure | EC$_{50}$ Results (µM) | |
|---|---|---|---|
| | | 72 h | 96 h |
| 1587 | | 0.91 | 0.94 |
| 1038 | | 9.41 | 7.27 |
| 1001 | | 3.90 | 3.68 |
| 1064 | | 5.27 | 4.65 |
| 1133 | | 4.54 | 3.76 |
| 1135 | | 21.15 | 18.78 |
| 1134 | | 25.18 | 25.15 |
| 1221 | | 6.90 | 7.03 |
| 1039 | | 8.56 | 10.60 |
| 1200 | | 7.03 | 6.29 |
| 1207 | | 5.42 | 4.98 |
| 1214 | | 14.02 | 10.95 |

**Fig 6. PBZ compounds used in this study and their EC50 values.**

presented an intriguing result as IPP treatment sensitized parasites to the inhibitor (Fig 8B and 8C). The reason(s) for IPP sensitization of PBZ1214 is unclear at present and will require further investigation.

## Discussion

Chaperonins (CPN60/ HSP60) function biochemically as ATP-dependent molecular chaperones, working in concert with other chaperones and co-chaperones to facilitate protein folding and maintain proteostasis. Chaperonins are originally prokaryotic proteins and are essential in almost all bacteria, where they are known as GroEL, as well as in all endosymbiotic organelles [23]. Indeed, an HSP60 homolog is found in the mitochondria of almost all eukaryotes, including human and *Plasmodium*, although the latter has not been well characterized. Another GroEL homolog is found in chloroplasts [38] (an essential Rubisco binding subunit) as well as in related non-photosynthetic plastids, including the apicoplast organelle of *Apicomplexan* parasites. As we show here, the apicoplast resident CPN60, is quite distinct from the *Plasmodium*

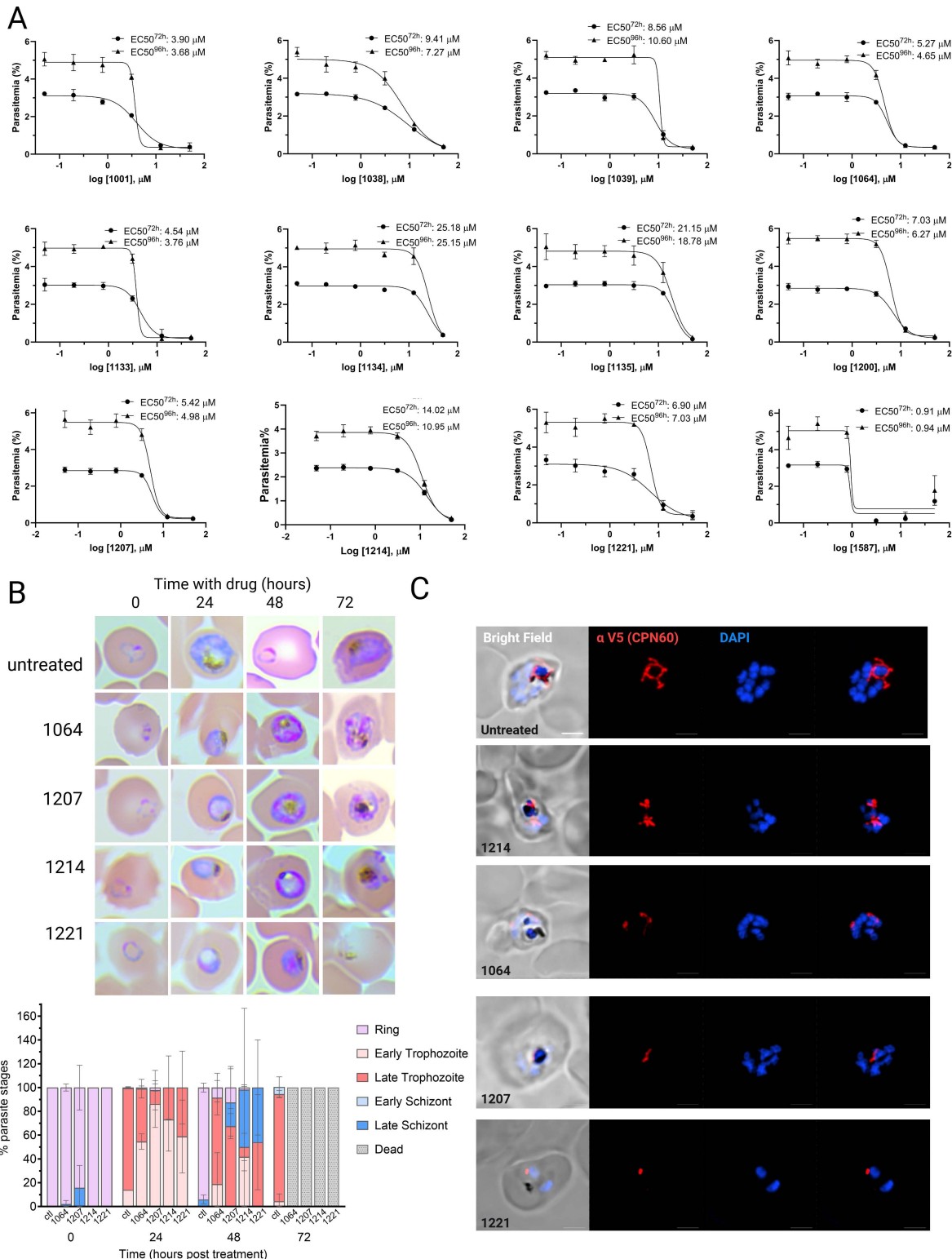

**Fig 7. Testing phenylbenzoxazole (PBZ) GroEL inhibitors for anti-*Plasmodium* activity.** A. *Wildtype* NF54 parasites were seeded at 0.5% parasitemia and incubated in serial dilutions of PBZ compounds, ranging from 50 μM to 50 nM. Parasitemia was measured after 72 or 96 hours to calculate the drugs' half-maximal effective concentrations (EC50). Data are fit to a dose-response equation and are represented as mean±SEM. B. CPN60[V5-apt]

parasites were synchronized and treated with 12.5 μM of different PBZ inhibitors. Up: Giemsa-stained blood smears were imaged using an upright Eclipse E200 Microscope. Bottom: Blood smears of CPN60[V5-apt] parasites with or without PBZ treatment were analyzed for specific parasites developmental stages. PBZ-treated parasites demonstrate a delayed trophozoite stage already on the first replication cycle, fail to egress, and die during schizogony. C. Immunofluorescence microscopy of CPN60[V5-apt] parasites following 24-hour incubation with 12.5 μM of different PBZ inhibitors. Z stack images processed as Maximum Intensity show from left to right: merged DIC contrast, anti-V5 antibody (CPN60, red), DAPI (parasite nucleus, blue), and merge of fluorescent channels. The imaging was performed using a Nikon Spinning Disk confocal fluorescence microscope equipped with a 100x/1.4NA objective. Scale bar is 2.5 μM.

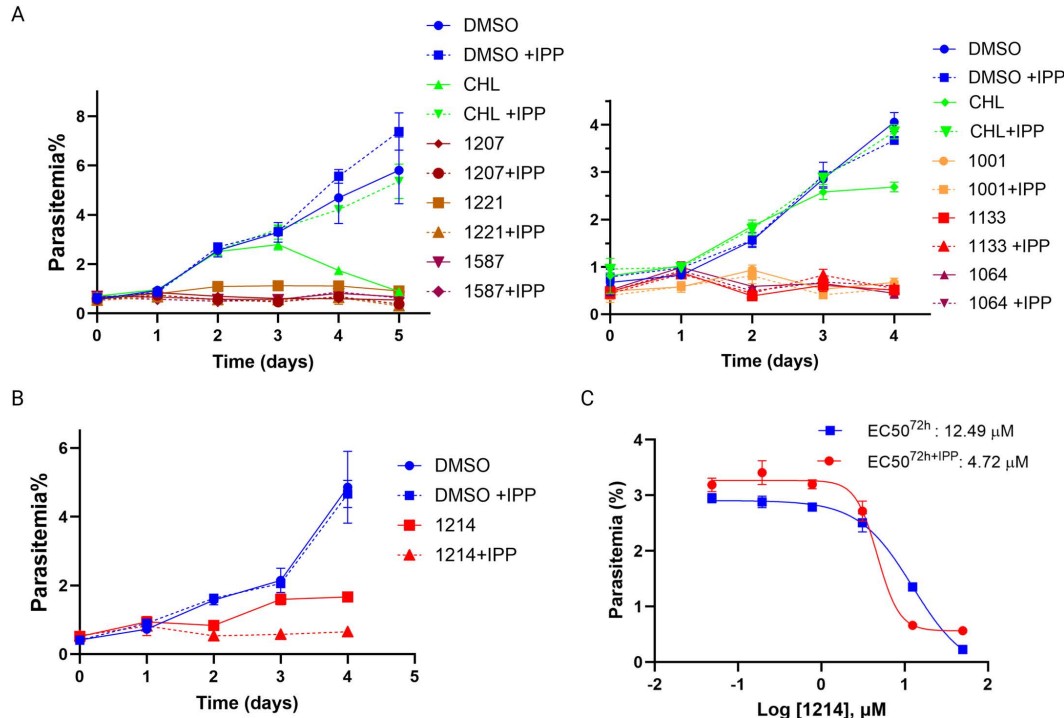

**Fig 8. Testing specificity of PBZ compounds against apicoplast activity.** A. *Wildtype* NF54 parasites were incubated with either DMSO (blue, control), 30μM Chloramphenicol (green, CHL) or 12.5μM of different PBZ compounds (shades of red), with (dashed) or without (solid) 200μM IPP. Parasitemia was monitored every 24 h over at least two cycles via flow cytometry. 100% of growth represents the highest value of calculated parasitemia (final parasitemia with DMSO). Normalized data are represented as mean ± SEM for three technical replicates. B. *Wildtype* NF54 parasites were incubated with DMSO (blue, control) or 12.5μM of PBZ1214 (red), with (dashed) or without (solid) 200μM IPP. Parasitemia was monitored every 24 h over two cycles via flow cytometry. Data are represented as mean ± SEM for three technical replicates. C. *Wildtype* NF54 parasites were seeded at 0.5% parasitemia and incubated in serial dilutions of PBZ1214 with (red) or without (blue) 200μM IPP. Parasitemia was measured after 72 hours to calculate the drugs' half-maximal effective concentrations (EC50). Data are fit to a dose-response equation and are represented as mean ± SEM.

mitochondrial Hsp60 which in turn shares a higher degree of conservation with human mtHSP60 and *E. coli* GroEL. Prior works in *Plasmodium* and in *Toxoplasma gondii* reported on the expression of an apicoplast CPN60 [26,39]. However, so far, the cellular functions of the apicoplast CPN60 have not been resolved.

In this work, we found that CPN60 expression profile correlates with the timing of apicoplast elongation, branching and division [29,40], which supports its newly-observed function in organelle biogenesis. Interestingly, CPN60 knockdown induced immediate cell-death within the first replication cycle. This phenotype is different and faster than the more typical apicoplast-delayed cell death, observed in parasites treated with apicoplast-targeting antibiotics [30]. This kinetics is more similar to the effect of drugs blocking isoprenoids synthesis, which also target the apicoplast but kill within one replication

cycle [11,41]. Thus, CPN60 knockdown adds to growing evidence of immediate cell death resulting from inhibition of api-coplast biogenesis [42–45], underlying its potential as a drug target.

In bacteria and plant chloroplasts, CPN60 is critical for heat shock (HS) response. In the case of *Plasmodium*, the parasite copes and responds to periodic fever episodes during infection of the human host [46,47]. Since the ER has a limited repertoire of molecular chaperones [48], it would have made sense if the apicoplast chaperonin complex complements the cellular stress response. Indeed, a recent large-scale forward-genetic screen suggested a link between the apicoplast and fever-survival [31]. However, we found that during recovery after HS, CPN60 levels remained constant, and that perturbating CPN60 levels did not affect HS response, most likely precluding a role for CPN60 in this stress.

In the apicoplast of *P. falciparum*, Clp proteins form an organellar degradation machinery that is independent of the cytoplasmic proteasome [15,22]. In previous studies [14,15], we showed that a tightly synchronized sequence of events determines the activity of the Clp complex. In the current work, we found that CPN60 interacts with both the inactive and active PfClpP protease, affecting its activation turnover. While these co-IP experiments demonstrate interaction in the native cellular environment, they cannot reveal whether the binding is direct or rather mediated through a third partner. Similar to *Plasmodium*, the chloroplast Clp system is the primary machinery for protein degradation in the plastid stroma [19]. A recent cryo-EM study describing the structure of the chloroplast Clp complex, found that CPN20 co-chaperonin, a co-factor of CPN60, forms a cap on the top of ClpP and is shown to repress ClpP proteolytic activity *in vitro* [49]. In *Plasmodium*, there are three potential CPN20-like genes that are predicted to localize to the apicoplast. However, none of the structural predictions that we tried could fit any of them into a stable interaction model with either PfClpP or CPN60. Thus, future experimental work will be needed to understand the nature of the Clp-CPN complex interaction and whether it is mediated by co-chaperonins or other factors.

In the absence of such information, we used our IP data and previous structural works to try and predict how a direct interaction between CPN and Clp complexes might look. A previous study solved the crystal structure of *Plasmodium* CPN60 [33]. Although in this study it was annotated as the *Plasmodium* mitochondrial HSP60, careful examination of the sequence used indicates that it is rather the structure of the apicoplast CPN60 (PF3D7_1232100). As seen in its solved structure, the basic oligomeric state of CPN60 is a single heptameric ring, though a double-ring tetradecameric structure is also possible (PDB ID 7K3Z, S3C Fig). Similarly, a solved X-ray structure of PfClpP demonstrates that it forms a tetrade-camer, consisting of two heptameric rings (PDB ID 2F6I, S3D Fig) [22]. Our AF3 predictions provided a model in which the PfClpP heptameric ring tilts one way, whereas the CPN60 ring tilts the other way (S3E Fig). In this configuration it is possible for both CPN60 and ClpP complexes to form tetradecameric barrels, which interact via a stable interface (See our working model in Fig 9A). The stability of this interaction can be assessed by computational alanine scanning that identifies potential interaction hotspots. This sort of analysis reveals significant hotspots in key residues in both complexes; The ClpP predicted hotspots residue are Y41, I45 and Y73 and the CPN60 hotspots residues are K243, F320 and N323 (Fig 9B). Additional experimental validation is required to confirm these interaction models and to refine our understanding of the molecular mechanisms driving the CPN60-ClpP complex formation. Interestingly, we discovered that upon CPN60 removal, active PfClpP levels decline, while PfClpP zymogen levels increase (Fig 9C). Until this point, we operated under the hypothesis that, similar to chaperonin orthologs in other organisms, the apicoplast CPN60 acts as a "conformational editor" to assist in organellar folding processes. These unexpected results showed that at least part of its activity is to stabilize and activate the Clp complex, thus supporting organellar degradation processes. Based on these results, we propose a working model in which the interdependence between two complexes with opposing functions, facilitates a balanced proteostasis in the apicoplast (Fig 9).

Over years of research, chaperonins have been identified as potential targets for the development of novel agents affecting their activity. Many of those have implications for new-generation antibiotics, cancer and neurodegeneration treatment, as well as the development of diagnostic biomarkers [23,50–52]. Nevertheless, the potential of mitochon-drial and/or apicoplast CPN60 as viable targets for antimalarial development have not been investigated. Here we

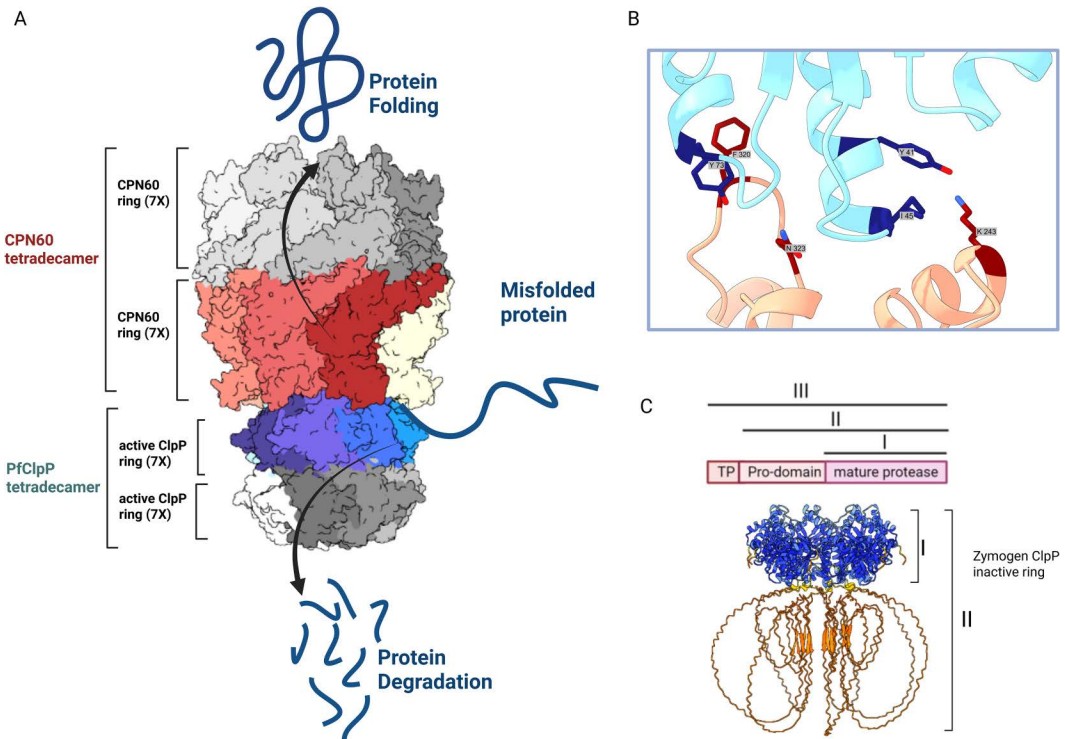

**Fig 9. A model for regulation of proteostasis in the apicoplast by CPN and Clp complexes.** A. Two opposing mechanisms interact and balance proteostasis in the apicoplast; The CPN complex consists of the ATPase chaperonin CPN60 which is arranged in two heptameric rings assisting in refolding organellar proteins. The Clp proteolytic complex in its core contains the PfClpP protease which is, likewise, arranged in two heptameric rings that degrade organellar proteins. The model shows a complete representation of the interaction between the two tetradecamers of CPN60 and PfClpP as predicted by AF3. PfClpP is depicted in blue and teal shades, while CPN60 is colored in red, brown, and beige. A hypothetical model of the full tetradecamers, constructed based on the solved structures, is shown in grey tones. PfClpP rings can be found in two forms; one as an inactive zymogen which contains an inhibitory pro-domain (shown in C) and a processed form representing the proteolytically active ring (shown in this model). Both forms bind the CPN complex, and the switch between the two forms enable a balance between degradation and folding and determine organellar proteostasis state. B. Alanine scanning predicting hotspot residues in the interaction between PfClpP and CPN60. CPN60 is shown in pink, and PfClpP in light blue. Key interaction residues ("hotspots") are highlighted in dark blue for PfClpP and dark red for CPN60. C. Structure prediction of the oligomerized PfClpP zymogen coloured by confidence levels according to pLDDT scores. High-confidence regions are shown in blue, while low-confidence regions are depicted in red, indicating the varying reliability of the predicted structure. The upper panel shows schematic representation of PfClpP posttranslational processing.

tested the antimalarial activity of a small panel of phenylbenzoxazole (PBZ) compounds that were originally identified as inhibitors of the *E. coli* GroEL/ES chaperonin system [36]. These inhibitors exhibited antibiotic effects from the low-µM to mid-nM range against a panel of Gram-positive and Gram-negative bacteria [50], mycobacteria [53] and even *Trypanosoma* parasites [54], highlighting the versatility of targeting chaperonins. Recently, Godek *et al.* demonstrated PBZ analogues that functioned on-target against GroEL in *E. coli* [37]. Within the group of PBZ anti-GroEL inhibitors tested here, we found compounds with antimalarial effect at the low and sub-micromolar range which affect apicoplast biogenesis. However, their cytotoxic effects were not reversed with IPP supplementation, suggesting additional targets beyond the apicoplast CPN60. This lack of specificity could be explained by the presence of two other chaperonin systems in the parasite in addition to the one in the apicoplast. One of them is the mitochondrial HSP60, that like the apicoplast CPN60 belongs to the Type I chaperonins which are common to all bacteria and endosymbiotic organelles in humans, plants and parasites. Another is the type II chaperonin, which is found in archaebacterial cytoplasm (thermosome) and eukaryotic cytosol (CCT/TRiC) [55]. Based on recognized and separated evolutionary

lines, the TriC system is generally poorly characterized comparing to the type I HSP60. Uncharacteristically, the CCT/TRiC system in *Plasmodium* has been investigated more than its type I variant, and shown to be a valid drug target [56–58]. Whether it is also targeted by PBZ inhibitors, or can explain the IPP toxicity that is associated with some of them, remains to be investigated.

To conclude, by investigating the relationship between the *Plasmodium* Chaperonin and Clp complexes, we revealed two opposing mechanisms that interact to balance proteostasis in the apicoplast. These detailed cellular, biochemical and bioinformatic analyses shed light on how excessive degradation is avoided when folding is compromised, to regain steady state. We further provide a foundation for systematic evaluation of these mechanisms and the potential of targeting CPN60 as an antimalarial strategy. Future studies focusing on their bacterial evolutionary origins could use existing antibacterial compounds to streamline drug discovery.

## Materials and methods

### Plasmids construction

Genomic DNA was isolated from *P. falciparum* using NucleoSpin blood kit (Machery-Nagel). For the generation of the CPN60[V5-apt]; conditional mutant, 500 bps from the C-term and 3'UTR from the of *cpn60* gene (Pf3D7_1232100) were amplified from genomic DNA using primers P1+P2 and P3+P4, respectively. The two products were conjugated together using overlap extension PCR and inserted as one piece into pMG74-tetR-Dozi vector [27]. For expression of a C-term CPN60 guide RNA, oligos P8+P9 were annealed and inserted into pUF1-Cas9-guide. Primer P7 was used in combination with primer P6 or P4 to test for accurate integration in the parasite genome following transfections. Evaluating aptamer repeats number was done using primers P5+P6.

For the generation of ClpP[DEAD] parasites, we transfected parasites with the puc-HSP110-ClpP[DEAD-Ty] plasmid which we used previously to study ClpP processing and activation [15]. It encodes the entire Open Reading Frame of ClpP (PF3D7_0307400), mutated in three residues: Ser264Ala, Glu308Arg and Arg285Glu, following by a triple Ty tag. Expression is driven under the *Pfhsp110c* gene (PF3D7_0708800). Briefly, the HSP110 expression system comprises of a repair plasmid that contains homology sequences from the *Pfhsp110c* gene. The repair sequences include the last 429 bp (not including the stop codon) from the *Pfhsp110c* gene, followed by a 2A skip peptide, ClpP[DEAD-Ty] sequence and the first 400 bps from *Pfhsp110c* 3'UTR. For expression of a *Pfhsp110c* guide RNA, oligos P10+P11 are expressed from pUF1-Cas9-guide. Primers P12 and P13 were used to test for accurate integration in the parasite following transfections.

All constructs utilized in this study were confirmed by Sanger sequencing. PCR products were inserted into the respective plasmids using In-Fusion Snap assembly Master mix (Takara). All restriction enzymes used in this study were purchased from New England Biolabs. Oligonucleotides used in this study are summarized in Table 1.

### Cell culture and transfections

Parasites were cultured in RPMI medium supplemented with Albumax I (Gibco) and transfected as described earlier [59–61]. Synchronization of cultures were achieved by intermittent sorbitol and Percoll treatments. To generate CPN60[V5-apt] parasites, a mix of two plasmids was transfected into NF54 *wildtype* parasites in the presence of 0.5 mM anhydrotetracycline (aTC, Cayman Chemicals). The plasmid mix contained 40 µg of pUF1-Cas9-guide expressing the CPN60 guide, and 20 µg of pMG74-based donor plasmid with CPN60 homology regions, linearized by EcoRV. Drug pressure was applied 48 hours post transfection using 2.5 µg/ml Blasticidin (BSD, Sigma). For each transfection, at least 2 clones (A4 and F11) were isolated via limiting dilutions and used in all subsequent experiments.

To generate CPN60[V5-apt]; ClpP[DEAD-Ty] parasites, a mix of two plasmids was transfected into CPN60[V5-apt] parasites. The mix contained 2 plasmids; 40 µg of pUF1-Cas9-*Pfhsp110c*-guide and 40 µg of the marker-less repair plasmid

**Table 1. Primers and Oligos used in this study.**

| Primers | Primer name | Primer sequence |
| --- | --- | --- |
| P1 | Apt-CPN60-Cterm-EcoRV-F | TTTTTTTTGGATATCGAATCTGAAATGGAATTACAAAAAATGGGGGCTAATATAG |
| P2 | CPN60 cterm apt R | TTGGTTTACCCTCGAGTTCATCATAATTGTATCCGTCGTTCATGCTATCTTCATCATC |
| P3 | Apt-AflII-CPN60–3UTR-F | CTTTCCGGGCGCGCCTTAAGCACACATATATATATATATATATATATAAATTG |
| P4 | CPN60–3UTR-EcoRV-Apt-R | TTCAGATTCGATATCCAAAAAAAACAAAAATTTTTATGTATTTATCCATATAC |
| P5 | aptamer Forward | CTTATGACGTACCTGATTATGCAC |
| P6 | aptamer Reverse | GTAGACCCCATTGTGAGTACATAAATATATTATATAAACTAGACTAGG |
| P7 | CPN60 integr specific F | GTT CAA ATA ATG ACG AAA AGA AAT ATC TTG AAC TAA TAG G |
| P8 | U6 CPN60 guide F | TAAGTATATAATATTGATGAAGATTCTATGAATGAGTTTTAGAGCTAGAA |
| P9 | U6 CPN60 guide R | TTCTAGCTCTAAAACTCATTCATAGAATCTTCATCAATATTATATACTTA |
| P10 | U6 HSP110 guide F | TAAGTATATAATATTGACAAACTTGGATGCAAATGGTTTTAGAGCTAGAA |
| P11 | U6 HSP110 guide R | TTCTAGCTCTAAAACCATTTGCATCCAAGTTTGTCAATATTATATACTTA |
| P12 | HSP110-Cterm pcr test F | GAGATCGTATTTTACTTTCCTTAGATGATTATGAAAATTGG |
| P13 | HSP110-UTR pcr test R | GCATCAATGCACAAAATAAAAATCATACAATTTGGGG |
| P14 | CPN60 qPCR F | GTCATTTTCATCTTTACATAATTTTCTTTTC |
| P15 | CPN60 qPCR R | TTTGTTGCTTGTAATATTTTTGTTATGTTTGCGG |
| P16 | Aldolase qPCR F | GCAAAGAGTATTATAAAGCTGGTGCAAGGTTTGC |
| P17 | Aldolase qPCR R | GCATATCTAGCCAATCCCCATGCAGTTTCGTG |
| P18 | arginine-tRNA ligase qPCR F | AAGAGATGCATGTTGGTC |
| P19 | arginine-tRNA ligase qPCR R | GTACCCCAATCACCTACA |
| P20 | LRR5 qPCR F | AATGTTTTGGACACGTTAATAGAAGAACAG |
| P21 | LRR5 qPCR R | CTACAGTATATTCCTTCTCATCAAAATCG |

puc57-HSP110-ClpP$^{DEAD}$. Drug pressure was applied 48 hours post transfection, using 1 µM DSM1 (BEI Resources), selecting for Cas9 expression. DSM1 was removed from the culturing media once parasites clones were isolated by limiting dilution. For each transfection, at least 2 clones (D12 and H10) were isolated via limiting dilutions and used in all subsequent experiments.

## Growth assays

For CPN60 knockdown assays, asynchronous parasite cultures were washed 8 times and incubated without aTC. Throughout the course of the experiment, parasites were sub-cultured to maintain parasitemia between 1–5% and parasitemia was monitored every 24 hours via flow cytometry. Growth assay during PfClpS knockdown was performed following the same steps. The parasitemia levels were determined in comparison to negative control (uninfected RBCs). Cumulative parasitemia at each time point was back calculated based on actual parasitemia multiplied by the relevant dilution factors. Parasitemia in the presence of aTC at the end of each experiment was set as the highest relative parasitemia and was used to normalize parasites growth. Normalized data from technical replicates were analyzed and fit to an exponential growth curve using Prism (GraphPad Software, Inc.) All experiments shown are representative of at least three biological replicates. For IPP rescue, media was supplemented with 200 µM IPP (Isoprenoids LC). Controls for apicoplast ablation were done using 30 µM chloramphenicol. For apicoplast ablation during CPN60 knockdown, parasites were incubated without aTC for 17 days while being supplemented with IPP. After these 17 days, this culture was divided into two groups, one with aTC (to allow CPN60 re-expression and accumulation) and one without aTC. Both groups were supplemented with IPP. After additional 16 days, each group was further divided into two cultures, with or without IPP. The growth of the four treated groups were then measured by flow cytometry, as well as analysed by IFA and WB.

## Flow cytometry

Aliquots of parasite cultures (5 µl) were stained with 8 µM Hoechst (ThermoFisher Scientific) in PBS. The fluorescence profiles of infected erythrocytes were measured and analyzed by flow cytometry on a CytoFlex S (Beckman Coulter). The parasitemia data were analyzed and presented using Prism 10.5.0 (GraphPad Software, Inc.).

## Recovery after heat shock

To induce heat shock response, parasites were incubated at 40°C for 24 hours and then allowed to recover at 37°C to mimic the characteristic malarial fever episodes. Growth of recovered parasites was monitored over the next following days using flow cytometry and protein samples were isolated as described below. When CPN60 levels were to be manipulated, aTC was first washed away, and parasites were subjected to heat shock 24 hours later to allow protein levels to drop.

## Half-maximal effective concentration (EC50)

To generate an EC50 curve for aTC, asynchronous cultures of CPN60$^{V5-apt}$ or CPN60$^{V5-apt}$; ClpP$^{DEAD-Ty}$ parasites were washed 8 times and then seeded at 0.5% parasitemia in a 96 well plate with varying concentrations of aTC. To measure aTC EC50 after recovery from heat shock, parasites were similarly treated, incubated for 24 hours at 40°C, and then returned to 37°C for the rest of the experiment. To generate EC50 curves for HSP60 inhibitors, NF54 *wildtype* parasites were seeded at 0.5% parasitemia in a 96 well plate with varying concentrations of each compound. In all cases parasitemia was measured after 72- and 96-hours using flow cytometry. Data were fit to a dose-response equation using Prism 10.5.0 (GraphPad Software, Inc.).

## In-house anti-Ty tag monoclonal antibody purification

The hybridoma BB2 clone cell line (a kind gift from Boris Striepen) was adapted to serum-free medium. 500 ml spent medium of the hybridoma clone containing mAbs was collected, concentrated and purified by HiTrap Protein G HP antibody purification 1 ml column (Cytiva) operated by peristaltic pump, as per manufacturer protocol. The bound antibody was eluted using 10 ml 0.1 M Glycine-HCl buffer, pH 2.7 for ten minutes, and then neutralized with 1M Tris-HCl, pH 9.2 buffer. The protein content was determined and adjusted to 1 mg/ml.

## Immunoprecipitation

Co-Immunoprecipitation (Co-IP) protocols were performed using anti-V5 antibody (CPN60$^{V5-apt}$) or anti-Ty antibody (ClpP$^{DEAD-Ty}$). Pellets from $4 \times 10^8$ parasites were isolated using cold saponin and were lysed and sonicated in Extraction Buffer (40 mM Tris HCl pH 7.6, 75 mM KCl, and 1 mM EDTA, 15% glycerol) supplemented with HALT protease inhibitor (Thermo). 10% of the sample was kept for later analysis (input sample). Rabbit anti-V5 (Cell Signaling, D3H8Q) or anti-Ty, BB2 (inhouse purification) antibodies were crosslinked to Dynabeads protein G beads (Invitrogen) by incubating with 5 mM BS3 crosslinker (CovaChem) for 30 minutes. Quenching of crosslinker was performed using 1 M Tris HCl (pH 7.5) for 30 minutes and then washing with 1.2M Glycine HCl (pH 2.5) to remove excess unbound antibody. Antibody-conjugated beads were then washed 3 times with PBS and incubated with the supernatant at 4°C. Washes were performed using a magnetic rack (Life Technologies). Samples were run on SDS-page and blotted with anti-V5 or anti-Ty antibodies.

## Western blot

Western blots were performed as described previously [15]. Briefly, parasites were collected and host red blood cells were permeabilized selectively by treatment with ice-cold 0.04% saponin in PBS for 15 min, followed by a wash in ice-cold PBS. Cells were lysed using RIPA buffer, sonicated, and cleared by centrifugation at 4oC. The antibodies used in this study were mouse anti-Ty, BB2 (in-house purification), rabbit anti-V5, D3H8Q (Cell Signaling, 1:3000), mouse anti-V5, E9H80 (Cell Signaling,

1:3000), rabbit anti-aldolase ab207494 (abcam, 1:3000). The secondary antibodies that were used are IRDye 680CW goat anti-rabbit IgG and IRDye 800CW goat anti-mouse IgG (LICOR Biosciences, 1:20,000). The Western blot images and quantifications were processed and analyzed using the Odyssey infrared imaging system software (LICOR Biosciences). Three biological replicates were used for quantification of protein expression levels using Prism 10.5.0 (GraphPad Software, Inc.).

## Microscopy and image processing

For IFA, cells were fixed using a mix of 4% paraformaldehyde and 0.015% glutaraldehyde and permeabilized using 0.1% Triton-X100. Primary antibodies used are mouse anti-Ty1, BB2 (Invitrogen, 1:100), and rabbit anti-V5, D3H8Q (Cell Signaling, 1:100). Secondary antibodies used are Alexa Fluor 488 and Alexa Fluor 546 (Life Technologies, 1:100). For mitochondrial staining, live cells were incubated with 300 nM Mito-tracker Deep Red (Invitrogen) for 15 minutes, and then fixed and stained as described above. Cells were mounted on Fluoroshield with DAPI (Sigma). The imaging was performed using a Nikon Spinning Disk confocal fluorescence microscope equipped with a 100x/1.4NA objective. Images were collected as Z-stack, and displayed as maximum intensity projection. Image processing, analysis and display were performed using Zeiss ZEN 3.7 Software. Adjustments of brightness and contrast were made for display purposes. Blood smears were imaged using an upright Eclipse E200 Microscope (Nikon), equipped with x100 oil objective Type NVH, and captured using a digital Sight 1000 microscope camera and NIS-Elements Software (Nikon).

## Ultrastructure expansion microscopy

Ultrastructure expansion microscopy was performed as previously described [62] with minor modifications. Briefly, parasite cells were adhered to poly-d-lysine coated coverslips, washed and fixated in PFA, and then incubated on gels, followed by transfer to denaturation buffer. Gels were then incubated at 95°C, and then expanded in doubled-distilled water. The gels were then measured for expansion factor (5X in this case). The gel is then blocked (3% BSA) and incubated with antibodies. Dyes and antibodies used in this study: rabbit anti-V5, D3H8Q (Cell Signaling, 1:100), Alexa Fluor 488 goat anti-rabbit (1:200, Cell Signaling), NHS-ester Alexa Fluor 405 (1:250, Thermo Fisher Scientific) and SYTOX Deep Red (1:1000, Thermo Fisher Scientific). The imaging was performed using Airyscan2 LSM980 confocal microscope equipped with 63x/1.4NA objective.

## RNA extraction and quantitative real time PCR (qRT-PCR)

Total RNA was extracted using the Macherey-Nagel NucleoSpin RNA Prep Kit. RNA concentration and purity were assessed using a NanoDrop spectrophotometer. For reverse transcription, 10 µL of the total RNA (from a 40 µL eluate) were used to synthesize complementary DNA (cDNA) using the QuantaBio qScript cDNA Synthesis Kit.

Quantitative PCR was performed using the Luna Universal qPCR Master Mix (New England Biolabs) on a QuantStudio 5 Real-Time PCR System. Gene-specific primers were used to amplify CPN60 (P14 + P15), LRR5 (PF3D7_1432400, P20 + P21), and the two housekeeping genes aldolase (P16 + P17) and arg-tRNA synthetase (P18 + P19) for normalization. Each reaction was performed in triplicate. Relative expression levels were calculated using the ΔΔCt method, normalizing to the geometric mean of the housekeeping genes and comparing expression levels in heat-shocked samples to the non-treated control. Statistical analyses were performed using Prism 10.5.0. (GraphPad Software, Inc.)

## AF3 structure prediction and Alanine scanning

The inputs are protein and peptide sequences. All different versions were run with random seed 2 or 1. Runs was done on alphafold server from GitHub (https://alphafoldserver.com/). Structural Visualization was done using the PyMOL Molecular Graphics System (Version 2.4.0) and UCSF ChimeraX (Version 1.4). Robetta computational alanine scanning was applied to define the critical interface residues, as implemented in http://robetta.org. Detailed protocol was described by Kortemme *et al* [59].

PLOS Pathogens

**Generating the apicoplast and mitochondrial PfCPN60 models with PBZ1587 bound at the ring-ring interface**

CryoEM structures were recently reported indicating PBZ1587 binds with 7-fold symmetry between adjacent subunits across the ring-ring interface of *E. coli* GroEL [37]. To support whether PBZ inhibitors would similarly bind to and inhibit the apicoplast and mitochondrial PfCPN60 isoforms, we generated homology models for the two PfCPN60 isoforms based on the GroEL-1587 cryoEM structure (9C0B) using Schodinger Maestro. We generated two models for each isoform, one with the apical domains in a conformation where the substrate binding sites are oriented inwards towards the central cavity (like in the GroEL-1587 9C0B structure), and the other with the apical domains in an extended conformation and oriented upwards from the central cavity (like in the apicoplast PfCPN60 structure), described as follows.

1. Apicoplast PfCPN60–1587 model with apical domains positioned inwards: We first employed the build homology model, multiple templates (homomultimer) function using default parameters and aligning the apicoplast PfCPN60 sequence onto the GroEL-1587 tetradecamer structure (9C0B). This provided a reasonable initial model for further refinements of the ligand binding sites. As the alpha-helices spanning residues 496–541 present in the apicoplast PfCPN60 structure were not represented as such in the homology model, we then spliced these secondary structures into the model. Next, we independently refined each of the 7 binding sites using the Refine Protein-Ligand function using default parameters, with the exception of refining atoms within 10 Å of the ligands. We then conducted Induced Fit Docking of the PBZ1587 inhibitor independently at each site using default parameters for Extended Sampling, with the exception of allowing refining of residues within 8.0 Å of ligand poses. The docked structure was then subjected to final refinement through the Protein Preparation Workflow using default parameters.

2. Mitochondrial PfCPN60–1587 model with apical domains positioned inwards: We first employed the build homology model, multiple templates (homomultimer) function using default parameters and aligning the mitochondrial PfCPN60 sequence onto the GroEL-1587 tetradecamer structure (9C0B). Next, we independently refined each of the 7 binding sites using the Refine Protein-Ligand function using default parameters, with the exception of refining atoms within 10 Å of the ligands. We then conducted Induced Fit Docking of the PBZ1587 inhibitor independently at each site using default parameters for Extended Sampling, with the exception of allowing refining of residues within 8.0 Å of ligand poses. The docked structure was then subjected to final refinement through the Protein Preparation Workflow using default parameters.

3. Apicoplast PfCPN60–1587 model with apical domains extended upwards: We first aligned the individual apicoplast PfCPN60 rings on those of the GroEL-1587 tetradecamer structure (9C0B) so as to retain the apical domains in the extended, upwards conformation as in the reported PfCPN60 structure. Next, we independently refined each of the 7 binding sites using the Refine Protein-Ligand function using default parameters, with the exception of refining atoms within 10 Å of the ligands. We then conducted Induced Fit Docking of the PBZ1587 inhibitor independently at each site using default parameters for Extended Sampling, with the exception of allowing refining of residues within 8.0 Å of ligand poses. The docked structure was then subjected to final refinement through the Protein Preparation Workflow using default parameters.

4. Mitochondrial PfCPN60–1587 model with apical domains extended upwards: We first employed the build homology model, multiple templates (homomultimer) function using default parameters and aligning the mitochondrial PfCPN60 sequence onto the apicoplast-1587 homology model generated above, with the apical domains extended upwards (model 3). Next, we independently refined each of the 7 binding sites using the Refine Protein-Ligand function using default parameters, with the exception of refining atoms within 10 Å of the ligands. We then conducted Induced Fit Docking of the PBZ1587 inhibitor independently at each site using default parameters for Extended Sampling, with the exception of allowing refining of residues within 8.0 Å of ligand poses. The docked structure was then subjected to final refinement through the Protein Preparation Workflow using default parameters.

## Supporting information

**S1 Fig.  Amino Acid sequence alignment between (1)** *E. coli* **GroEL (UNIPROT P0A6F5), (2)** *Plasmodium falciparum* **apicoplast CPN60 (PF3D7_1232100), (3)** *Plasmodium falciparum* **mitochondrial Hsp60 (PF3D7_1015600), and (4) human mitochondrial Hsp60 (UNIPROT P10809).** Green represents Identity and high similarity (>80%) and Yellow similarity (>60%). Note the N-terminal extension of CPN60 representing the transit peptide which is removed upon apicoplast localization. Additional extensions appearing only in CPN60 are two inner loops and the long C-terminal stretch, which may interfere with PBZ binding.
(DOCX)

**S2 Fig.   A.** PfClpS$^{apt}$ parasites were cultured in the presence or absence of aTC and IPP, and parasitemia was measured every 24 hours for 7 days using flow cytometry. Parasites lacking aTC display a growth defect beginning on day 4, followed by a decline in parasitemia. The addition of IPP rescues the growth defect, inidicating apicoplast dysfunction. Parasitemia was normalized to the maximum value observed under aTC treatment on day 7, and Normalized data are represented as mean±SEM for three technical replicates. **B.** CPN60$^{V5-apt}$ parasites were washed and incubated with different aTC concentrations (8 nM, 4 nM, 16 nM and 500 nM), subjected to HS and then allowed to grow at 37°C for three days, while being measured daily by flow cytometry. Data were fit to an exponential (Malthusian) growth curve (graph shown on Fig 4F) and the doubling time as calculated is shown here.
(DOCX)

**S3 Fig.   A.** Genotyping by PCR to confirm ClpP$^{DEAD-Ty}$ integration at the *hsp110* locus. Genomic DNA was purified from transfected isolated parasites clones (D12 and H10), and primers P12 and P13 were used to specifically amplify the integrated region. A shift of 1000 bp corresponds to the integration of ClpP$^{DEAD-Ty}$. **B.** Co-IP of CPN60$^{V5-apt}$. Parasites were isolated and sonicated, and extracts were incubated with anti-V5 antibody-conjugated beads (for CPN60 pulldown) Input and IP samples were loaded on SDS-page and blotted with anti-Ty, anti-V5 antibodies and anti-aldolase as a negative control. **C.** The X-ray structure of apicoplast CPN60 (PDB ID: 7K3Z) [33]. The different chains are highlighted in different yellow and red colors, revealing the heptameric arrangement of the CPN60 ring. On the left is a top view, and on the right is a side view. **D.** The X-ray structure of PfClpP (PDB ID: 2F6I) [22]. The different chains are highlighted in different blue shades, revealing the heptameric arrangement of the PfClpP ring. On the left is a top view, and on the right is a side view. **E.** AF3 structure prediction of the interaction between the two heptameric rings of CPN60 and PfClpP. The PfClpP ring is shown tilting upward, binding to the upper side of CPN60. PfClpP is depicted in blue and teal shades, while CPN60 is coloured in red and beige. On the left is a side view, and on the right is a top view.
(DOCX)

**S4 Fig.   Overall GroEL/CPN60/HSP60 and ring-ring interface binding site residue conservation.** Residues that were observed to interact with ligand in the GroEL: PBZ-1587 cryoEM structure are shown in grey, with corresponding residues from sequence alignments for *P. falciparum* apiCPN60, *P. falciparum* mtCPN60, and human mtHSP60 shown below. Conserved residues are represented as dots. Percent identical (ID) and similar (Sim) residues for the ring-ring interface and overall chaperonin sequences are shown to the right.
(DOCX)

**S5 Fig.   A.** Residue: ligand interaction map as predicted from analysis of the apiCPN60: PBZ-1587 and mtCPN60:PBZ1587 homology models. Color-coding indicates the degree of interaction: Orange = residue more distant with no appreciable interaction; Yellow = residue adjacent but no significant interaction; Light Green = positive interaction with corresponding residue from one ring; Dark Green = positive interactions with corresponding residues from both rings. **B.** Images of PBZ-1587 bound in the apiCPN60 and mtCPN60 homology models.
(DOCX)

**S6 Fig.** Overlay of the apiCPN60: PBZ-1587 and mtHSP60: PBZ-1587 homology models. The mtCPN60 rings are shown as grey solvent exposed surfaces (with two individual subunits across the ring-ring interface shown in yellow), the apiCPN60 protein backbones are shown as red tubes, and bound PBZ-1587 inhibitors are shown as green surfaces. Unlike *P. falciparum* mtCPN60 and chaperonin orthologues from other species, apiCPN60 has additional loop residues extending from the apical domains (residues 370–392), and extended alpha-helices at the ring-ring interface (residues 502–529).
(DOCX)

**S7 Fig.** Testing phenylbenzoxazole (PBZ) GroEL inhibitors during partial CPN60 knockdown. A. CPN60[V5-apt] parasites were washed eight times and the incubated with 8nM aTC for 24 hours. The following days these parasites were seeded at 0.5% parasitemia and incubated in serial dilutions of PBZ compounds, ranging from 50 μM to 50 nM. Parasitemia was measured after 72 or 96 hours to calculate the drugs' half-maximal effective concentrations (EC50). Data are fit to a dose-response equation and are represented as mean ± SEM.
(DOCX)

**S1 Data.** Raw data used to generate article figures.
(XLSX)

## Acknowledgments

We thank Boris Striepen for the BB2 hybridoma cell line; Florentin lab members for critical reading and comments on the manuscript; Yael Feinstein-Rotkopf at the Imaging Core Facility at the Faculty of Medicine, En Kerem; VEuPathDB and PlasmoDB for scientific, academic and community support.

## Author contributions

**Conceptualization:** Amanda Tissawak, Anat Florentin.

**Formal analysis:** Amanda Tissawak, Steven M. Johnson, Anat Florentin.

**Funding acquisition:** Anat Florentin.

**Investigation:** Amanda Tissawak, Yarden Rosin, Shirly Katz Galay, Alia Qasem, Michal Shahar, Anat Florentin.

**Methodology:** Amanda Tissawak, Yarden Rosin, Shirly Katz Galay, Alia Qasem, Michal Shahar, Nirit Trabelsi, Ora Furman-Schueler, Steven M. Johnson, Anat Florentin.

**Project administration:** Anat Florentin.

**Resources:** Nirit Trabelsi, Ora Furman-Schueler, Steven M. Johnson.

**Supervision:** Anat Florentin.

**Validation:** Amanda Tissawak, Anat Florentin.

**Writing – original draft:** Amanda Tissawak, Anat Florentin.

**Writing – review & editing:** Amanda Tissawak, Michal Shahar, Nirit Trabelsi, Ora Furman-Schueler, Steven M. Johnson, Anat Florentin.

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
