## [Decision Letter · Decision Letter 0]

A Chaperonin Complex Regulates Organelle Proteostasis in Malaria Parasites

PLOS Pathogens

Dear Dr. Florentin,

Thank you for submitting your manuscript to PLOS Pathogens. After careful consideration, we feel that it has merit but does not fully meet PLOS Pathogens's publication criteria as it currently stands. Therefore, we invite you to submit a revised version of the manuscript that addresses the points raised during the review process.

Please submit your revised manuscript within 60 days Apr 11 2025 11:59PM. If you will need more time than this to complete your revisions, please reply to this message or contact the journal office at plospathogens@plos.org. Please include the following items when submitting your revised manuscript:

We look forward to receiving your revised manuscript.

Kind regards,

Tracey J. Lamb

Section Editor

PLOS Pathogens

Tracey Lamb

Section Editor

PLOS Pathogens

Editor-in-Chief

PLOS Pathogens

orcid.org/0000-0003-2946-9497

Michael Malim

Editor-in-Chief

PLOS Pathogens

orcid.org/0000-0002-7699-2064

**Journal Requirements:**

At this stage, the following Authors/Authors require contributions: Amanda Tissawak, Yarden Rosin, Michal Shahar, Nirit Trabelsi, Ora Furman-Schueler, Johnson Steven M, and Anat Florentin. Please ensure that the full contributions of each author are acknowledged in the "Add/Edit/Remove Authors" section of our submission form.

2) Please note that the Author Summary should appear in your manuscript between the Abstract and the Introduction.

- TM on page: 19.

5) Please include a full list of legends for your Supporting Information files after the references list.

6) We note that your Data Availability Statement is currently as follows: "All of the data described are included in the manuscript and Figures." Please confirm at this time whether or not your submission contains all raw data required to replicate the results of your study. Authors must share the “minimal data set” for their submission. PLOS defines the minimal data set to consist of the data required to replicate all study findings reported in the article, as well as related metadata and methods (https://journals.plos.org/plosone/s/data-availability#loc-minimal-data-set-definition).

7) Please amend your detailed Financial Disclosure statement. This is published with the article. It must therefore be completed in full sentences and contain the exact wording you wish to be published.

3) If any authors received a salary from any of your funders, please state which authors and which funders.

**Reviewers' Comments:**

Reviewer's Responses to Questions

**Part I - Summary**

Reviewer #1: This paper investigates the biological role of the apicoplast-targeted CPN60 chaperonin in P. falciparum parasites. In a prior study, they identified an interaction between CPN60 and the apicoplast ClpP protease. In the present study, they demonstrate that knockdown of CPN60 is lethal to parasites. This lethality does not seem to involve an altered sensitivity to heat stress but does involve loss of the mature ClpP. Structural modeling suggests a possible role for CPN60 in stabilizing ClpP and contributing to maintenance of the apicoplast proteome.

Although functional annotation of apicoplast-targeted proteins has identified many factors proposed to contribute to protein homeostasis in this key organelle, few of these proteins have been directly studied, including CPN60. The present study thus adds valuable insight to understanding protein quality-control mechanisms in the apicoplast. Overall, I am enthusiastic about the contribution of this manuscript to general apicoplast understanding. There are a few conclusions that seem overstated and/or would be strengthened by additional controls or analyses, but these critiques do not detract from an overall positive appraisal.

Reviewer #2: This manuscript describes a series of genetic, parasitological and bioinformatic studies of CPN60, a Plasmodium falciparum chaperonin that the authors previously localised to be an apicoplast resident protein that interacts with the Clp complex. In this work they establish convincingly that it plays an essential role in P. falciparum intraerythrocytic development, and that this essential function is carried out in the apicoplast, as it can be rescued using IPP supplementation. They go on to argue that CPN60 is not involved in heat shock response, and that instead it regulates the Clp proteolytic complex, but the data for this is less compelling, and the structural predictions lack some level of rigour or validation. Overall this is a substantive and novel piece of work, but significant issues need to be addressed if the central findings are to be substantiated.

Reviewer #3: Group I chaperonins (Cpn60) constitute a well-studied family of chaperoning in bacteria (GroEL/GroES) and endosymbiotic organelles (mitochondria, chloroplast and Plasmodium apicoplast). Th major role is preventing protein aggregate formation by competing with misfolding and aggregation. A Cpn60 member was previously identified in the apicoplast of Toxoplasma gondii and expected to perform its usual functions in preventing protein misfolding and assisting protein import (Ref. 26). The authors present a compressive characterization of PfCPN60. The quality of the data is good and most interpretations are convincing. Together, this study is an important contribution to a better understanding of apicoplast functions.

**Part II – Major Issues: Key Experiments Required for Acceptance**

Reviewer #1: 1. Figure 1D/E and lines 141-149: The authors conclude discordance between temporal expression of CPN60 mRNA versus protein.

a. Did the authors determine mRNA levels in the edited NF54 line? They cite ref. 38, which suggests the mRNA levels in Fig. 1D are from the prior published study in 3D7 and HB3. This comparison between different strains and WT versus edited CPN60 genes seems problematic, as any discordance could be due to these strain and/or genetic differences.

b. The limited temporal resolution in the WB analysis in Figure 1E weakens a conclusion of meaningful lag in protein levels from mRNA levels. It is possible that protein levels also increase by 24 hours, which the current data cannot rule out.

2. Figure 5: The co-IP data does not specify if association between CPN60 and ClpP is direct or indirect. Are there biochemical precedents or other considerations that suggest that this association is likely direct? If not, the authors should explicitly note this uncertainty before proceeding to model as a direct interaction. In this regard, the AF modeling is speculative, which is fine to include as long as acknowledged.

3. The Discussion largely repeats observations presented in the Results sections. Can the authors hone the Discussion to focus on considerations that go beyond re-stating results? These considerations might include possible additional functions/interactions for CPN60, evolution of Cpn60 function in plastids, or prior precedents for collaborative function between CPN60 and ClpP systems.

Reviewer #2: 1) Is CPN60 involved in Heat Shock response? Two lines of evidence are used to argue that it is not – firstly, the expression level of CPN60 does not increase after incubation of parasites at 40C for 24 hours, and secondly minor reductions in CPN60 protein level via manipulation of expression using TetR do not affect the parasite response to heat shock. In both assays the recovery time at 37C after heat shock is considerable, raising concerns that a short-term effect is being missed and therefore making it impossible to make a definitive statement that CPN60 is not involved in heat shock. More worryingly, there is no positive control included in either assay – ie a protein that does increase in expression post-heat shock, or whose depletion affects the heat shock response, making the suitability of the assays impossible to predict. Either the experiments in Fig 3 all need repeating with a clear positive control, and/or a wider range of conditions needs to be explored and the claims about the potential role of CPN60 in heat shock need to be considerably down-regulated.

2) AlphaFold predictions. A sizeable section of the manuscript centres on AlphaFold predictions of the putative interaction between CPN60 and Clp complex. There is however no quantitative assessment of the strength of these predictions, making their significance difficult to assess, nor is there any functional validation of the predicted interaction interface. This section needs a much more robust analysis (rather than quite generic statements such as that a certain complex structure is more “plausible”) and ideally validation of interacting residues.

3) Changes in ClpP abundance following CPN60 depletion. This is really the only validation of the AF predictions, but there are experimental issues with the data. As the authors show very elegantly in Figure 2, depletion of CPN60 results in parasites that die at the end of the first developmental cycle, ie within 48 hours. There are therefore many things going on in CPN60 depleted parasites, and a decrease in ClpP abundance is therefore not necessarily directly linked to CPN60 depletion. In addition, the quantitation approach is unclear – there is still clearly CplP present at 72 hours (despite the data in Fig 2G suggesting all parasites at this time are ‘dead schizonts’), but no quantitation for this timepoint is shown in Fig 5H. Where the bands at all time points normalised to aldolase or other markers to account for overall changes in protein abundance following CPN60-induced death? Surely a more direct test of the AF models would be to use targeted mutagenesis of predicted interacting residues in their tagged catalytically dead ClpP expression system? There is a possibility that expression of such mutants could have a dominant negative phenotype, but the lack of an impact of additional expression of catalytically dead ClpP gives some hope that this may not be the case, and it would be a much more direct test of their bioinformatic AF predictions.

4) Specificity of PBZ compounds. The approach of using inhibitors of bacterial GroEL proteins is an interesting one, but the lack of IPP rescue (other than a subtle effect for inhibitor 1214) suggests that the anti-parasitic impact of these inhibitors is very likely not only due to anti-CPN60 activity. Without a clear link of these inhibitors to CPN60, the focus of the rest of the manuscript, the data is not compelling. Perhaps the authors could try a drug interaction approach, ie using the TetR-CPN60 line to see if there is a synergistic or antagonistic interaction between CPN60 depletion using ATc and the PBZ compounds? If these compounds are acting at least in part via CPN60, then depleting CPN60 could in theory alter the EC50 of the PBZ compounds by altering their abundance of their target. Compound 1214 would seem the most obvious target, given the potential partial IPP rescue?

Reviewer #3: Figure 2: I accept the idea that knockdown of CPN60 results in an earlier arrest (at the trophozoite stage) than caused by drug-induced delayed death. However, the tetR-system is very slow, and a side-by-side comparison with another knockdown mutant (e.g. in the apicoplast ribosome) will formally be required to substantiate the claim. This is clearly beyond the scope of the study, but perhaps toning down the interpretation is warranted.

Figure 3: In the absence of a positive control (any Plasmodium HSP) the claim about a missing heat-shock response remains somewhat untested. I would like to see a HSP upregulation to be convinced that under the conditions a heat shot response was tested properly.

Figure 4: Since the colocalization to the apicoplast is an important point, additional life cycle stages and a mitochondrial marker will be useful.

Figure 5 (and corresponding section on pages 8 and 9): Panels B-F are not primary data and should be moved to supplemental material. The results section can be shortened. This will also make the interesting data shown in Fig. 5G/H more accessible.

Figures 6/7: the inhibitor studies are very nice. I was expecting a life cycle analysis similar to Fig. 2G to further substantiate the claim of apicoplast CPN60 being the target. Will that be possible? As is and given the lack of IPP rescue, the explanation that a mitochondrial chaperonin is targeted by the compounds (lines 348 & 349) is the most plausible.

**Part III – Minor Issues: Editorial and Data Presentation Modifications**

Reviewer #1: 1. Lines 135-138 and Figure 1C: What is the impact of apicoplast disruption (e.g., doxycycline/IPP) on CPN60 processing? Do the lower bands disappear if apicoplast import is perturbed?

2. Lines 159-161: Delayed death most commonly observed for translation inhibitors, as the authors note. It is well established that other types of apicoplast dysfunctions distinct from translation inhibition cause more rapid parasite death (e.g., PMIDs 29109165, 33135634, 30674649). The authors may wish to modify this text to clarify that delayed death is not a general property of all apicoplast dysfunctions or inhibitors.

3. Figure 2: Does loss of CPN60 cause apicoplast disruption (assessed by microscopy and/or PCR analysis of the apicoplast genome) in the presence of IPP? This prediction seems to be a strong expectation of CPN60 knockdown in the presence of IPP rescue but is not tested by the authors.

4. Figure 5: Is the pulldown of CPN60 with ClpP due to a specific or promiscuous interaction? The presence of multiple background bands in the WB, weak CPN60-V5 signal, and lack of negative control(s) make it difficult to assess specificity of this interaction.

5. Figure 1E: Is the V5 signal normalized to the aldolase signal? I assume so but normalization is not described explicitly.

6. Figure 2A: Based on known DNA-binding properties of TetR, the RNA-aptamer binding region of TetR is expected to differ from the aTC binding pocket (PMID 8153629), in contrast to the depiction in this scheme.

7. The details of synchronization for the experiment in Figure 1C are not given. Are they the same as given in the legend of Figure 2? Can authors include these details in the Methods section?

8. Figure 3: This data would be easier to compare if the authors combined panels D and E into a single plot with both datasets colored differently.

9. Line 185: “compared to the its levels”. Remove “the”.

10. Figure 3H: It is difficult to follow the overlaid data. To simplify the comparison, the authors might consider fitting the data to an exponential growth model and to compare the rate constants for exponential growth to clarify growth impacts of heat stress with varying aTC concentrations.

11. Figure 4E: Can the authors quantify co-localization using a Pearson co-efficient (or similar analysis)?

12. Figure 5A: Inclusion of MW marker positions would be helpful.

13. Figure 7: The authors did not observe parasite rescue by IPP from PBZ inhibitor treatment, but is there evidence (e.g., by microscopy) for impaired apicoplast biogenesis? Compounds may still hit the apicoplast in addition to other targets.

14. Line 412: “hippomorphic” ;The authors presumably mean “hypomorphic”.

Reviewer #2: 1) Fig 2B – why is only one CPN60 band visible, rather than two in Fig 1C which the authors suggest is the result of post-translational modification?

2) How exactly is a ‘dead schizont’ defined morphologically? Scoring based on parasite morphology is sensitive to operator bias, but this category seems particularly hard to define and therefore hard to quantify accurately. The growth curves make the phenotype clear, but parasite stage scoring is less clear and it is not obvious how many parasites were counted in each time point.

3) Co-localisation of CPN60 and ClpP is compelling, but some attempt at quantification based on multiple images, rather than showing a single presumably ‘perfect’ image, would be a much more rigorous analytical approach.

4) Why was the approach taken to integrate a catalytic-dead version of ClpP at another genomic location under a heterologous promoter rather than simply tagging the endogenous gene using CRISPR-Cas9? Heterologous expression is always a risk in P. falciparum as it can cause expression and hence localisation artefacts. This is not a major concern in this case given the known location of the Clp complex, but some more detailed justification of the approach would have been helpful other than “to avoid adverse effects”.

Reviewer #3: Figure 3B/C: I probably overlooked something, but I don’t understand the data at the 48 h timepojnt. Blots are done in triplicate, so they all seem to miss any signal for either PfCPN60 or aldolase. Can the author provide an explanation?

Figure 4E: The immunofluorescence imaging is nice but I can’t see anything in the DIC image (schizont maybe?). Just omit or show other example.

Figure 5 G/H: Shouldn’t the x-axis labeling correspond (either 24, 48, 72 or 0, 24, 48)? Or alternatively the 0h time point should be displayed. The signal at 24 h is very weak and difficult to interpret one way or the other.

PLOS authors have the option to publish the peer review history of their article (what does this mean? ). If published, this will include your full peer review and any attached files.

**Do you want your identity to be public for this peer review?** For information about this choice, including consent withdrawal, please see our Privacy Policy .

Reviewer #1: No

Reviewer #2: **Yes: ** Julian Rayner

Reviewer #3: No

**Figure resubmission:**

**Reproducibility:**



---

## [Decision Letter · Decision Letter 1]

PPATHOGENS-D-24-02650R1

A Chaperonin Complex Regulates Organelle Proteostasis in Malaria Parasites

PLOS Pathogens

Dear Dr. Florentin,

Thank you for submitting your manuscript to PLOS Pathogens. After careful consideration, we feel that it has merit but does not fully meet PLOS Pathogens's publication criteria as it currently stands. Therefore, we invite you to submit a revised version of the manuscript that addresses the points raised during the review process.

Please submit your revised manuscript within 30 days Aug 01 2025 11:59PM. If you will need more time than this to complete your revisions, please reply to this message or contact the journal office at plospathogens@plos.org. Please include the following items when submitting your revised manuscript:

We look forward to receiving your revised manuscript.

Kind regards,

Tracey J. Lamb

Section Editor

PLOS Pathogens

Sumita Bhaduri-McIntosh

Editor-in-Chief

PLOS Pathogens

orcid.org/0000-0003-2946-9497

Michael Malim

Editor-in-Chief

PLOS Pathogens

orcid.org/0000-0002-7699-2064

**Journal Requirements:**

**Reviewers' Comments:**

**Part I - Summary**

Reviewer #1: The authors have thoughtfully revised the manuscript to address the critiques in the prior reviews. There are a few minor issues noted below that the authors should ideally revise. This manuscript substantively advances basic understanding of apicoplast protein quality control systems and will be a valuable addition to current knowledge.

Reviewer #2: The authors have responded comprehensively to the reviews. The new experimental data/controls added, plus the careful rewording of the interpretation in appropriate places, have together made the manuscript more robust and compelling. This is now definitely an important new step forward for understanding of apicoplast function, with interesting drug development implications, and thoroughly deserving of publication. Congratulations to the team.

Reviewer #3: The authors did an outstanding job to address all points raised. This reviewer is truely impressed by the commitment of the researchers to further substantiate the important findings of this work. I congratulate the authors - very convincing and interesting work with attention to detail!

**Part II – Major Issues: Key Experiments Required for Acceptance**

Reviewer #1: No major issues remain.

Reviewer #2: N/A

Reviewer #3: All issues were convincingly solved

**Part III – Minor Issues: Editorial and Data Presentation Modifications**

Reviewer #1: 1. Abstract (lines 25-26): I’m unsure what “traditionally accepted model” the authors refer to here. Delayed death has been reported most frequently for putative apicoplast translation inhibitors. For clarity, the authors may wish to rephrase as “deviating from the delayed death phenotype commonly observed for apicoplast translation inhibitors.”

2. Abstract (lines 31-21): For clarity and since a direct interaction is not uniquely established, the authors should consider qualifying as “A computational structural model of a possible direct interaction between these two large complexes…”

3. Abstract (lines 35-36): “with a likely additional target being the mitochondrial CPN60 isoform.” This line is speculative and thus odd to include in the abstract. I suggest removing it (although appropriate to mention in the Discussion).

4. Line 56: Apicomplexa is a phylum and not a family.

5. Lines 167-168: “An apicoplast-associated damage typically results in a delayed cell death”. This statement is misleading as there are multiple examples of non-translation inhibitors that target the apicoplast and cause first-cycle death (e.g., fosmidomycin and actinonin, see Ref. 44). For accuracy, the authors might rephrase as “Delayed cell death due to apicoplast-associated damage is observed in parasites treated with doxycycline or chloramphenicol.”

6. Lines 184-186: Supporting metabolic function and supporting apicoplast biogenesis are not mutually exclusive functions (e.g., SUF and MEP pathway activities are required for apicoplast biogenesis via IPP synthesis).

7. Lines 263-264: I’m unclear what the authors mean by “the cytoplasmic full length PfClpP fraction (III)”. Proteins targeted to and processed within the apicoplast matrix are expected to be co-translationally targeted to the ER before routing to the apicoplast, most likely within vesicles. For clarity, the authors might remove “cytoplasmic”.

8. Line 266: “real”, Do the authors mean “specific”? The co-IP results in this section do not distinguish direct versus indirect. For transparency, the authors should acknowledge this uncertainty before proceeding to assume and model as a direct interaction.

9. Line 333: “anti-Plasmodial”. The authors may wish to consult this reference (PMID: 22738856) on appropriate use of Plasmodium.

10. Figure 6C: Please label the channels in the figure (currently just described in the legend).

11. Figure 7C: Please color-code which label corresponds to which data curve.

Reviewer #2: N/A

Reviewer #3: None

PLOS authors have the option to publish the peer review history of their article (what does this mean? ). If published, this will include your full peer review and any attached files.

**Do you want your identity to be public for this peer review?** For information about this choice, including consent withdrawal, please see our Privacy Policy .

Reviewer #1: **Yes: ** Paul Sigala

Reviewer #2: No

Reviewer #3: No

**Figure resubmission:**
---

## [Editor Report · Decision Letter 2]

Dear Dr. Florentin,

We are pleased to inform you that your manuscript 'A Chaperonin Complex Regulates Organelle Proteostasis in Malaria Parasites' has been provisionally accepted for publication in PLOS Pathogens.

Best regards,

Tracey J. Lamb

Section Editor

PLOS Pathogens

Tracey Lamb

Section Editor

PLOS Pathogens

Sumita Bhaduri-McIntosh

Editor-in-Chief

PLOS Pathogens

orcid.org/0000-0003-2946-9497

Michael Malim

Editor-in-Chief

PLOS Pathogens

orcid.org/0000-0002-7699-2064
---

## [Editor Report · Acceptance letter]

Dear Dr. Florentin,

We are delighted to inform you that your manuscript, " A Chaperonin Complex Regulates Organelle Proteostasis in Malaria Parasites ," has been formally accepted for publication in PLOS Pathogens.

Best regards,

Sumita Bhaduri-McIntosh

Editor-in-Chief

PLOS Pathogens

orcid.org/0000-0003-2946-9497

Michael Malim

Editor-in-Chief

PLOS Pathogens

orcid.org/0000-0002-7699-2064